# BuildArena: A Physics-Aligned Interactive Benchmark of LLMs for Engineering Construction

**Tian Xia** [1 2] **Tianrun Gao** [1] **Wenhao Deng** [1] **Long Wei** [1] **Xiaowei Qian** [1] **Chenglei Yu** [1] **Tailin Wu** [1 2]

## Abstract

Engineering construction automation aims to transform natural language specifications into physically viable structures, requiring complex integrated reasoning under strict physical constraints. While modern LLMs possess broad knowledge and strong reasoning capabilities that make them promising candidates for this domain, their construction competencies remain largely unevaluated. To address this gap, we introduce **BuildArena**, the first physics-aligned interactive benchmark designed for language-driven engineering construction. Technically, it contributes to the community in two aspects: (1) an extendable task design strategy spanning static and dynamic mechanics across multiple difficulty tiers; (2) a 3D Spatial Geometric Computation Library for supporting construction based on language instructions. On nine frontier LLMs and three additional open-weight models, **BuildArena** comprehensively evaluates their capabilities for language-driven and physics-grounded construction automation. We release the code at https://github.com/AI4Science-WestlakeU/BuildArena to benefit construction automation in engineering applications.

## 1. Introduction

Engineering construction automation is an important field of AI for Engineering. It has various applications in domains such as automotive, transportation, and civil infrastructure (Wang et al., 2005; Domingues et al., 2016; Lin et al., 2019). The goal is to translate high-level task descriptions into executable end-to-end build plans that cover design, fabrication, and assembly. An ideal workflow lets users describe what

[1]Westlake University [2]Uniforce AI. Correspondence to: Tailin Wu <wutailin@westlake.edu.cn>.

*Proceedings of the 43rd International Conference on Machine Learning*, Seoul, South Korea. PMLR 306, 2026. Copyright 2026 by the author(s).

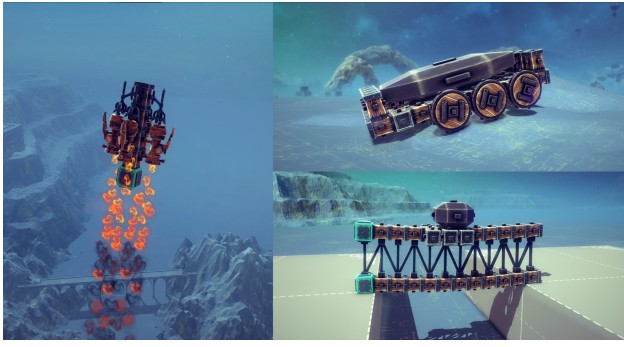

*Figure 1.* Examples of **BuildArena**'s construction results by LLMs, covering three tasks: **Lift** (left subfigure), **Transport** (upper right), and **Support** (lower right). All structures presented in the paper are built by LLMs.

they want in plain terms. For example, users request "*Design a rocket that meets Mars mission requirements.*" The system then creates realistic parts with precise material details and manufacturing specs. The workflow also provides assembly instructions that can be integrated into production systems. Such automation capabilities promise significant improvements in engineering efficiency and productivity.

Language-driven automated construction presents challenges in two key aspects. On one hand, it necessitates physics simulation environments with high fidelity to real-world constraints so that virtual designs and assembly procedures adhere to geometric, physical, and structural constraints. While modern physics engines and robotics benchmarks offer robust simulation capabilities (Todorov et al., 2012; Coumans & Bai, 2016; Makoviychuk et al.), there remains a gap in environments that integrate physics verification with language-driven multi-component assembly processes. On the other hand, the domain demands multi-level reasoning across long temporal horizons and 3D spatial contexts, as engineering artifacts inherently exhibit hierarchical organization, and their assembly follows sequential dependencies with strict feasibility constraints (Jiménez, 2013; De Fazio & Whitney, 2003). These factors collectively require both breadth of domain knowledge and depth of analytical thinking, challenging even for expert human engineers.

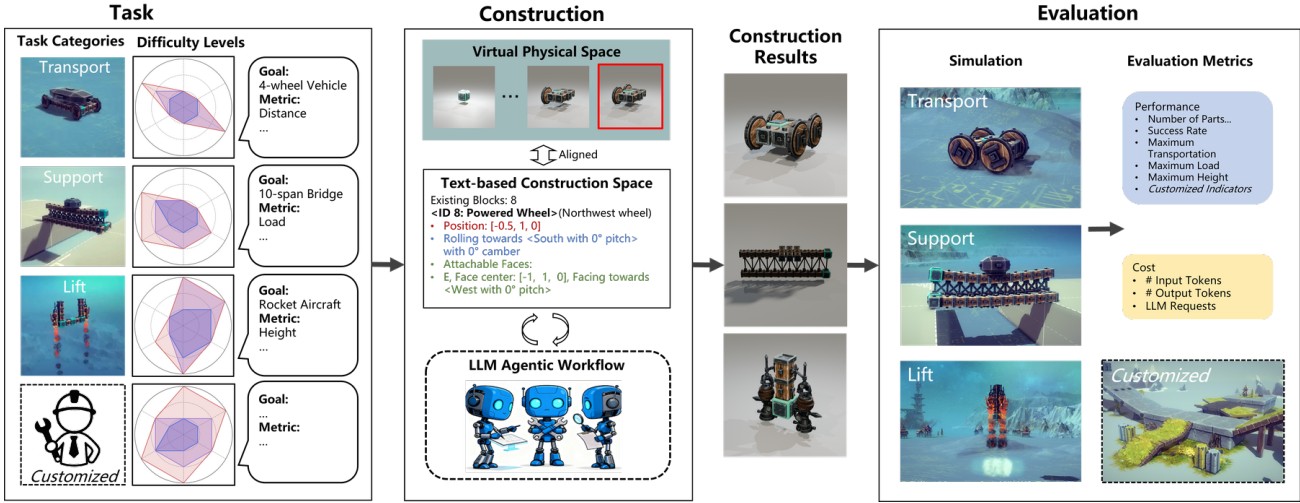

*Figure 2.* Illustration of our **BuildArena** framework. It contains three parts: (1) Task definition; (2) LLM-based Construction; (3) Simulation-based Evaluation. The arrows represent our pipeline. Components in dashed boxes, *i.e.*, task type, LLM agentic workflow, and simulator, could be customized by users. Details of the construction procedure is shown in Figure 4.

*Table 1.* Comparison between **BuildArena** and previous benchmarks.

| Benchmarks | Spatial Reasoning | 3D Construction | Construction-aimed Planning | Physical Simulator | Interactive Environment |
|---|:---:|:---:|:---:|:---:|:---:|
| PlanBench (Valmeekam et al., 2023) | ✗ | ✗ | ✗ | ✗ | ✗ |
| PlanQA (Rodionov et al., 2025) | ✓ | ✗ | ✗ | ✗ | ✗ |
| PHYRE (Bakhtin et al., 2019) | ✓ | ✗ | ✗ | ✓ | ✓ |
| VOYAGER (Wang et al., 2024a) | ✓ | ✗ | ✓ | ✓ | ✓ |
| Embodied Agent Interface (Li et al., 2024) | ✓ | ✓ | ✗ | ✓ | ✓ |
| **BuildArena (ours)** | ✓ | ✓ | ✓ | ✓ | ✓ |

Large language models (LLMs) have progressed rapidly in recent years, accumulating broad world knowledge and demonstrating strong capabilities in language understanding, mathematical reasoning, and code generation (Brown et al., 2020; Guo et al., 2025; Shao et al., 2024; Roziere et al., 2023). Moreover, they have shown proficiency in following human instructions, generating plans, invoking tools, and composing executable programs to interact with the external environments (Yao et al., 2023; Schick et al., 2023; Qin et al., 2024). These general intelligent capabilities position LLM agentic systems as promising candidates for automatic engineering construction.

Despite these advances, current evaluations on LLMs provide insufficient evidence of their capacity to construct physical entities. Established LLM benchmarks predominantly assess mathematical and programming capabilities (Cobbe et al., 2021; Hendrycks et al., 2021; Tang et al., 2025), which are evaluated mainly in textual or static environments, without interactions with physical environments. Existing physical reasoning datasets focus on physics understanding, but neglect the multi-step construction processes (Bakhtin

et al., 2019; Cherian et al., 2025). Meanwhile, research in programmatic 3D or CAD generation has advanced generation performance but rarely validates whether the generated designs yield executable assemblies under realistic physical conditions (Jones et al., 2020; Mallis et al., 2025). This interdisciplinary gap highlights the absence of frameworks to evaluate whether LLMs can effectively translate natural language specifications into physically viable assemblies. This limitation motivates our **research question**: *How can we comprehensively evaluate LLMs for language-driven and physics-grounded construction automation?*

In this paper, we answer the research question by proposing **BuildArena**, a physics-aligned interactive benchmark designed to assess LLMs' capabilities in engineering construction tasks. To our knowledge, except one concurrent work: BesiegeField (Zhang et al., 2025b) (see Appendix A for more details), this is the first benchmark that enables LLMs to perform 3D structure construction via natural language instructions and evaluates their performance within a physically constrained environment. **BuildArena** consists of three components: task definition, LLM-based construction,

and simulation-based evaluation, enabling in-depth comparison and analysis of LLMs. It supports customization of each component (see Figure 2). Examples of construction results are shown in Figure 1.

Comparison between **BuildArena** and existing benchmarks are provided in Table 1 (*See Appendix A for more related work*), implying that our work takes a first step towards construction-stage spatial assembly under physics constraints using LLMs. Thus, it substantially expands the scope of current LLM benchmarks to 3D construction domains. Our technical contributions are summarized as follows.

**We create an extensible task design strategy**. The strategy defines three task categories with quantifiable difficulty levels and corresponding evaluation metrics, serving as a reusable template for adding new categories and levels beyond the current instantiation.

**We develop a key framework module: a 3D Spatial Geometric Computation Library**. The library uses 3D computations and feedback in the iterative construction to ensure accurate execution of LLMs' language instructions. As the geometric computations in the widely used Besiege simulator (Spiderling, 2018) are closed-source and inaccessible, our open-source library reproduces its building operations.

## 2. Method

This section details our benchmarking methodology, including task setup in Section 2.1, language-driven and physics-grounded construction in Section 2.2, the LLM agentic workflow in Section 2.3, and the evaluation methodology in Section 2.4. Our method is illustrated in Figure 2.

### 2.1. Task

To design tasks in a principled manner, we first abstracted a set of difficulty dimensions that are commonly encountered in engineering practice:

**Quantification:** Extent of explicit numerical reasoning required (Olu-lawal et al., 2024).

**Robustness:** Tolerance to single-point failures (Geng et al., 2025).

**Magnitude:** Structural scale in span, load, and module count (Fan et al., 2023).

**Compositionality:** Required depth of hierarchical substructure construction and integration (Thurairajah et al., 2023).

**Precision:** Strictness of geometric requirement for placement and orientation (Gang et al., 2024).

**Ambiguity:** Clarity and completeness of task guidance (Moon et al., 2025).

Upon these dimensions, we construct three representative engineering task categories: **Support** (static structural stability), **Transport** (dynamic horizontal movement), and **Lift** (dynamic vertical movement). Within each task category, we defined three levels of difficulty, Easy (Lv.1), Medium (Lv.2), and Hard (Lv.3), by adjusting task details so that the corresponding requirements align with the above dimensions (see Figure 3). This design ensures both diversity of engineering scenarios and systematic coverage of engineering difficulty dimensions. Task descriptions, performance indicators and evaluation criteria of these three tasks are as follows. Detailed task content as LLM prompts are provided in Appendix H.

**Transport** focuses on constructing a machine capable of directional movement on a planar surface. It examines the LLM agents' ability to exploit the spatial movement afforded by given components. As the difficulty increases beyond Lv.1, the explicit instruction of building a four-wheeled vehicle is removed, and the transportation target changes from the machine itself into a cargo load with level-specific size, adding challenges on both instruction interpretation and building larger structures. The **Maximum Transport Distance** is chosen as the performance indicator, and we use a distance threshold as the criteria to identify if a machine is able to deliver effective transportation.

**Support** requires constructing a structure to support a load across a gap, aiming to test the ability of LLM agents to design and build bridges. The span of the gap multiplies across three levels, as larger span directly requires larger scale of the bridge, and makes the stable support harder. While only one structure is allowed in Lv.1, both Lv.2 and Lv.3 permit the modular construction with no more than three substructures without any detailed instruction, which also requires more precise assembly. We select **Maximum Load Weight** as the performance indicator for bridges, and use a minimum threshold to determine if a bridge successfully supports the load.

**Lift** requires constructing a rocket. At Lv.1, LLMs are explicitly required to build a single rocket engine without

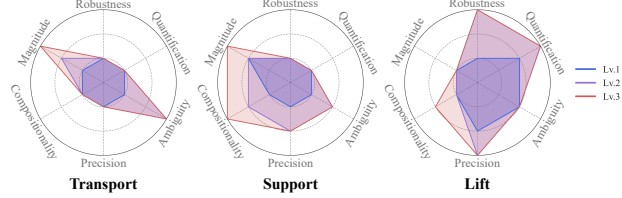

*Figure 3.* Difficulty profiles of the three task **Transport**, **Support**, and **Lift** across six engineering dimensions: Quantification, Redundancy, Scale, Modularity, Precision, and Ambiguity. Each radar chart illustrates how difficulty escalates from Lv.1 (blue) to Lv.2 (purple) and Lv.3 (red).

instruction on how to build, with **Thrust-to-Weight Ratio** (TWR) as the performance indicator. TWR $> 1$ represents the feasibility of providing effective thrust and marks successful construction. At Lv.2, the task requires LLMs to construct a rocket-powered aircraft as an integrated single structure. At Lv.3, LLMs must first build two separate substructures (a rocket engine and a support frame) before assembling the two into an aircraft. Both Lv.2 and Lv.3 tasks require that the aircraft are capable of launching from the ground. **Maximum Height** is adopted as the performance indicator, and the aircraft must reach a specific elevation to meet the success criterion. The escalation from Lv.1 to Lv.3 compounds multiple sources of engineering difficulty: higher demands on precise module alignment, the presence of multiple single points of failure (engine placement, structural balance), and strict requirements on modular construction and assembly. Together, these factors make `Lift` the most challenging task category among all three.

We note that the higher-difficulty tasks already exhibit long-horizon compositional structure: `Support` Lv.2/3 and `Lift` Lv.3 require planning substructures, constructing components, and assembling them into a final solution, taking up to hundreds of building actions per instance with iterative environment feedback during construction.

**Task Customization**. Task specifications can be customized as textual prompts that include task objectives, constraints, testing procedures, evaluation metrics, and any additional user-defined preferences to fully describe the task. The LLM agentic workflow (see Section 2.3) takes the specification as its input, generates candidate designs and carries out the construction steps. And the constructed structures are finally loaded into the simulation environment for evaluation.

### 2.2. Language-driven and Physics-grounded Construction

From the perspective of human engineering practice, construction is inherently an incremental and constraint-driven process. Structures are assembled step by step, each new component must be connected to existing ones, and physical feasibility (*e.g.*, collision avoidance) is continuously verified (Wilson & Latombe, 1994; De Fazio & Whitney, 2003). Each successful action requires accurate reasoning about the spatial relationships between new and existing structures. These features necessitate *Besiege* (Spiderling, 2018), an ideal platform to evaluate the LLMs for physics-grounded construction automation. Besiege is a popular construction sandbox game with realistic physics simulation, widely validated by world-wide player community to align with human physical intuition. It has a rich module space, a complete collection of basic structural and functional module types that can be combined to build complex objects, all completed by iteratively attachment and refinement of the native

modules (see Appendix F.1 for details about modules).

However, Besiege only provides graphical representation of 3D structures in the interactive construction space for human users, instead of natural-language illustration for LLMs. It also only supports direct manipulation through physical controller inputs, without any interface for symbolic or language-based interaction, nor any indirect API for programmatically operating the construction process. Therefore, we develop an *open-source Spatial Geometric Computation Library that faithfully mirrors Besiege's closed-source construction logic and physical constraints*, enabling LLMs to interact with the construction space through language interfaces. It ensures consistency between the effects of actions executed by LLMs in the language space and those performed by human users in the graphic interface, as illustrated in Figure 4. A quantitative fidelity validation on a 49-block machine shows that discrepancies between our library and Besiege are negligible (position error $< 1.5 \times 10^{-6}$ unit length, orientation error $< 2.5 \times 10^{-5}$ degrees; see Appendix F.5).

In implementation, it accepts an action announcing the operation and arguments from the LLM agent, computes and updates the state accordingly, and conducts physical constraint checks: it either returns a human-interpretable description of the current state, or prohibits the invalid action if constraints are violated and returns explanation of the failure. All actions fall into four categories *Build*, *Refine*, *Query*, and *Control*, detailed in Appendix F.2.

### 2.3. LLM agentic workflow

LLMs execute the construction procedure via LLM agentic workflow (Zhang et al., 2025a). For clear comparison between different LLMs, we restrict our consideration to workflows where all its entities employ the same LLM, differentiated by their respective prompts, as illustrated in Figure 2. We provide a fixed, shared workflow as a baseline evaluation protocol so that all models are compared under the same agentic setting, without model-specific tuning. Its design follows a coarse-to-fine structure (Xue et al., 2024) with multi-turn revision (Du et al., 2024), employing five entities: `Planner` (P), `Drafter` (D), `Reviewer` (R), `Builder` (B), and `Guidance` (G). In addition, `Controller` (C) is used for the task **Transport**. Prompt details are provided in Appendix I. The workflow includes three stages, as shown in the bottom row of Figure 4:

**Plan Phase**: Executed by `Planner`, this phase takes the task description and initial module set as input, outputting a structured construction plan in a predefined format.

**Draft–Review Loop**: Based on the generated plan, `Drafter` produces design schematics. `Reviewer` reviews and verifies the schematics, guiding `Drafter`'s re-

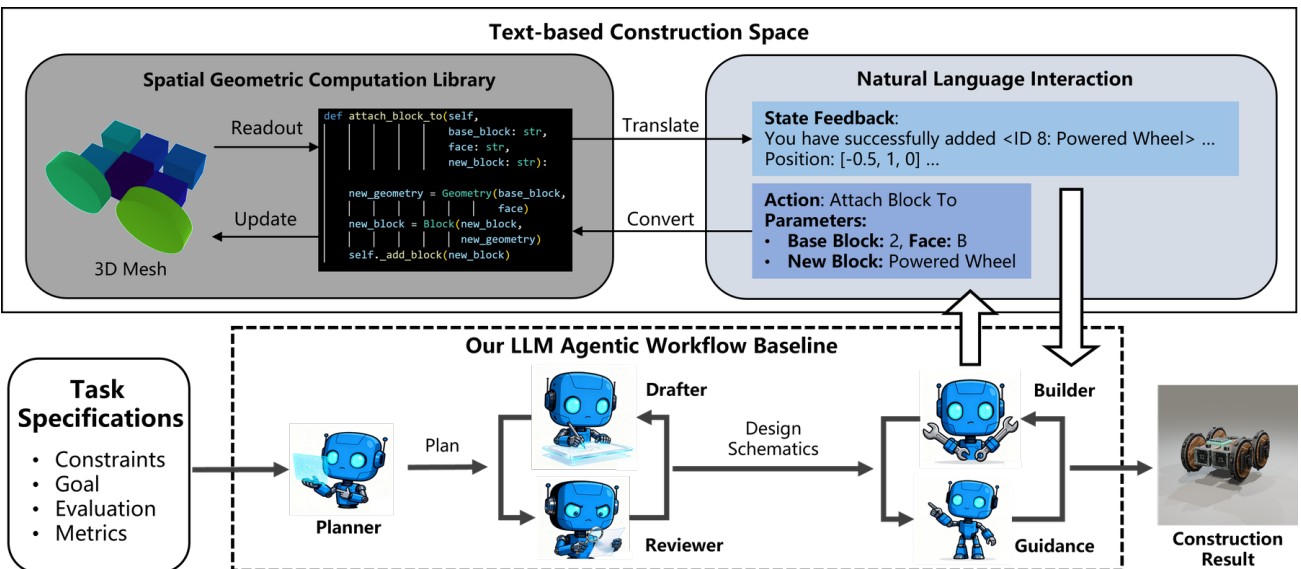

*Figure 4.* Details of the construction procedure in Figure 2. Our designed workflow (bottom row) contains five collaborative LLM entities, serves as a baseline for future user-customized alternatives. The text-based construction space (top row) has two transferable formats: code for physics-aligned spatial geometric computation, and natural language for LLM interface compatibility.

visions. The loop repeats until approval; and terminates in failure if the plan violates predefined rules.

**Build–Guidance Loop**: With approved schematics as input, `Builder` and `Guidance` collaborate on execution suggestions, building actions, and feedback. `Guidance` generates high-level suggestions step by step based on the draft, specifying the next action to invoke; `Builder` converts them into formatted construction commands for the Spatial Geometric Computation Library, which updates the states and returns either descriptive feedback or error prompts. The loop ends when `Guidance` confirms full completion, with the final state converted via the library into a conflict-free, simulation-compatible runnable file as the final result. Rejection by `Guidance` based on predefined rules terminates the process in failure. This workflow serves as a baseline for the development of advanced workflows in the future. The workflow component can be substituted with any user-customized one.

### 2.4. Evaluation

Our evaluation strategy is as follows. For each task-LLM pair, run the aforementioned construction procedure to produce a result (*e.g.*, a rocket) with detailed logs (*e.g.*, token consumption, conversation turns). Once construction is complete, the library exports the final structure into a Besiege-compatible file. A unified automation script then loads this file into the Besiege simulator, executes the task-specific simulation protocol, and records the resulting trajectories and metrics. To enhance reliability, the procedure is sampled 64 times for each task-LLM pair, with final reported results averaged over these 64 runs. Prompt and instructions for all tested LLMs are the same.

**Simulation environments are based on the Besiege sandbox**, with task-specific simulation protocols. For **Transport** tasks, the system evaluates if a constructed machine achieves effective motion. The LLM agent must specify a control configuration and sequence—invalid or missing controls cause immediate failure. The machine is then loaded into the environment. If repeated attempts show no effective movement, the system concludes the structure lacks mobility. For **Support** tasks, the environment provides fixed obstacles with varying gap widths according to difficulty level. A payload of gradually increasing weight is placed on the structure, and the simulation measures whether the machine can support and stabilize the load without collapse or loss of balance. For **Lift** tasks, Lv.1 records water cannons' heating status to calculate TWR; Lv.2–3 continuously activates water cannon firing to simulate launch, with module trajectories and cannons' heating status recorded for evaluation during a fixed window.

**Simulation environments can also be customized according to user-defined tasks**. Specifically, users set test conditions for the constructed result (*e.g.*, placing a load module above the constructed bridge) and configure simulation parameters, including the initial position, tracking points, and control information. All configurations are implemented by invoking our Spatial Geometric Computation Library. Finally, **BuildArena** executes the simulation via a unified script and collects log data throughout the entire process.

**Evaluation metrics cover performance and cost**. **Performance** includes three metrics: (1) Number of parts, referring

*Table 2.* Average ($n = 64$) performance comparison on different tasks across task levels Lv.1 (easy), Lv.2 (medium), and Lv.3 (hard). The indicator means maximum displacement for **Transport**; maximum load for **Support**; TWR for Lv.1 and maximum height for Lv.2, Lv.3 of **Lift**. The best results are in **bold** and the second-best are underlined.

| Task | Model | Number of Parts | | | Success Rate (%)↑ | | | Indicator↑ | | |
|---|---|---|---|---|---|---|---|---|---|---|
| | | Lv.1 | Lv.2 | Lv.3 | Lv.1 | Lv.2 | Lv.3 | Lv.1 | Lv.2 | Lv.3 |
| **Transport** | GPT-5 | **15.0** | **54.2** | **82.7** | **78.1** | **23.4** | **26.6** | 30.7 | **14.6** | 16.9 |
| | GPT-4o | 11.0 | 13.1 | 18.4 | 9.4 | 1.6 | 7.8 | 13.5 | 4.2 | 5.1 |
| | Claude-4 | 9.5 | 13.4 | 26.1 | 17.2 | 4.7 | 15.6 | **34.9** | 6.8 | 7.3 |
| | Grok-4 | 12.0 | 15.6 | 34.6 | 25.0 | 0.0 | 9.4 | 19.6 | 4.2 | 12.3 |
| | Gemini-2.0 | 9.0 | 12.3 | 10.0 | 1.6 | 1.6 | 1.6 | 4.8 | 4.6 | 4.6 |
| | DeepSeek-3.1 | 9.2 | 11.6 | 16.7 | 6.2 | 0.0 | 1.6 | 6.2 | 4.6 | 4.6 |
| | Qwen-3 | 9.1 | 7.8 | 10.9 | 10.9 | 1.6 | 4.7 | 18.4 | 3.3 | **21.5** |
| | Kimi-K2 | 14.2 | 17.1 | 19.5 | 12.5 | 0.0 | 1.6 | 7.4 | 4.5 | 3.5 |
| | Seed-1.6 | 7.4 | 11.2 | 28.5 | 6.2 | 4.7 | 7.8 | 4.3 | 3.8 | 6.8 |
| **Support** | GPT-5 | **40.1** | **80.3** | **115.6** | **85.9** | **59.4** | **10.9** | **324.9** | **178.4** | **24.9** |
| | GPT-4o | 36.8 | 16.7 | 29.9 | 40.6 | 0.0 | 0.0 | 181.2 | 0.0 | 0.0 |
| | Claude-4 | 8.0 | 21.4 | 31.5 | 7.8 | 1.6 | 0.0 | 36.8 | 3.3 | 0.0 |
| | Grok-4 | 18.7 | 22.3 | 33.3 | 46.9 | 15.6 | 0.0 | 211.4 | 44.5 | 0.0 |
| | Gemini-2.0 | 20.5 | 23.5 | 41.0 | 23.4 | 0.0 | 0.0 | 105.5 | 0.0 | 0.0 |
| | DeepSeek-3.1 | 19.0 | 10.5 | 17.8 | 25.0 | 0.0 | 0.0 | 122.6 | 0.0 | 0.0 |
| | Qwen-3 | 18.6 | 18.0 | 22.2 | 12.5 | 4.7 | 0.0 | 70.5 | 13.9 | 0.0 |
| | Kimi-K2 | 23.3 | 34.7 | 16.5 | 29.7 | 4.7 | 0.0 | 122.6 | 18.6 | 0.0 |
| | Seed-1.6 | 33.4 | 36.2 | 68.8 | 45.3 | 9.4 | 3.1 | 197.4 | 25.8 | 7.1 |
| **Lift** | GPT-5 | 5.2 | 8.0 | **15.6** | **95.3** | 10.9 | **17.2** | 4.3 | 127.8 | **366.2** |
| | GPT-4o | 4.0 | 7.4 | 5.1 | 7.8 | 3.1 | 0.0 | 0.9 | 9.1 | 4.1 |
| | Claude-4 | 4.5 | 7.9 | 2.5 | 10.9 | 1.6 | 0.0 | 1.0 | 4.1 | 1.2 |
| | Grok-4 | 4.8 | 6.8 | 1.1 | 31.2 | **31.2** | 3.1 | 1.8 | **890.6** | 86.5 |
| | Gemini-2.0 | 4.3 | 6.3 | 4.9 | 0.0 | 0.0 | 0.0 | 0.5 | 2.8 | 0.8 |
| | DeepSeek-3.1 | 4.5 | 6.9 | 1.1 | 10.9 | 0.0 | 0.0 | 1.0 | 3.3 | 0.6 |
| | Qwen-3 | 3.5 | 7.2 | 3.2 | 3.1 | 0.0 | 0.0 | 0.6 | 2.7 | 0.8 |
| | Kimi-K2 | **5.3** | **15.1** | 12.0 | 6.2 | 3.1 | 0.0 | 0.7 | 44.8 | 1.8 |
| | Seed-1.6 | 3.5 | 3.3 | 0.0 | 6.2 | 0.0 | 0.0 | 0.9 | 1.7 | 0.0 |

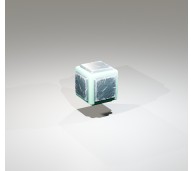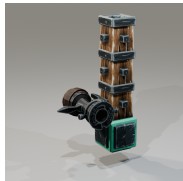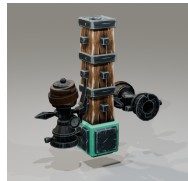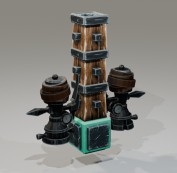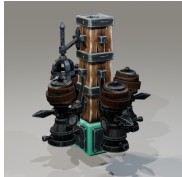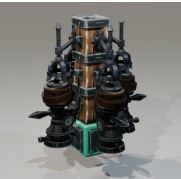

*Figure 5.* Example of the construction process. The rocket is constructed by Grok-4 for the **Lift** task under the Lv.2 (Medium) difficulty level. More examples are presented in Figure 10.

to the count of modules present in the construction result, reported as a descriptive signal of construction complexity rather than a ranking criterion. (2) Success rate, defined as the proportion of trials that successfully passed the criteria among 64 samples, a higher value is better. (3) Performance indicator, a task-specific metric extracted from simulation data that evaluates the performance under realistic physical conditions. A higher value is preferable for all indicators. Model ranking is determined solely by success rate and indicator via rank aggregation across tasks. Detailed suc-

cess criterion and indicator setup of each task are specified in Section 2.1. **Cost** is evaluated using three metrics: (1) number of accumulated input tokens, (2) number of output tokens, and (3) total number of LLM requests. A lower value is preferable for all the cost metrics.

## 3. Experiments

In the experiments, we aim to answer the following two questions: (1) Whether **BuildArena** serves as an effec-

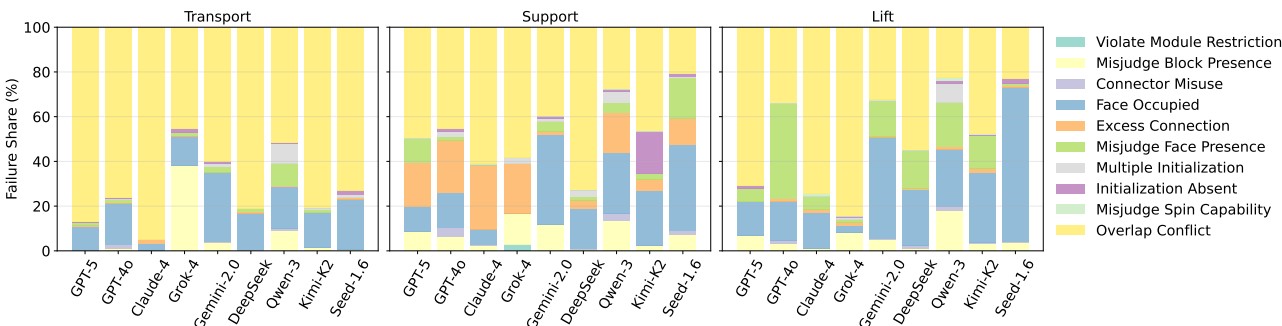

*Figure 6.* Distributions of failure reasons averaged over different LLMs (see Appendix E for detailed definitions of failure reasons).

tive benchmark for testing the construction capabilities of LLMs? (2) How current models perform within the **BuildArena** framework? To answer these questions, we evaluate nine frontier LLMs in **BuildArena**, including GPT-5, GPT-4o, Claude-4, Grok-4, Gemini-2.0, DeepSeek-3.1, Qwen-3, Kimi-K2, and Seed-1.6, and additionally test three open-weight models (Qwen3.5-9B, Qwen3.5-27B, Ministral-14B; see Appendix D). All simulations are conducted on Besiege. Model snapshots, module space, and simulation details are provided in Appendix G. We provide the **code** of **BuildArena** in this link.

### 3.1. Effectiveness of **BuildArena**

The performance of nine frontier models on **BuildArena** is presented in Table 2, with examples of construction results shown in Figure 9 and examples of construction process illustrated in Figure 10. These results demonstrate that, supported by the **BuildArena** evaluation framework, LLMs achieve language-based 3D construction automation, as evidenced by the following aspects. (1) Regarding task design, the diversity and difficulty levels are reasonably configured. Across individual tasks, performance tends to decrease as difficulty increases. An exception is the **Lift** task, where Lv.1 uses different metrics from Lv.2/3, making direct comparisons inappropriate. Specifically, at the Hard difficulty level of three tasks, most models exhibit low performance, yet a small number outperform others, indicating that the difficulty and criteria settings possess good discriminative power. (2) Concerning the LLM agentic workflow, numerous successful construction outcomes validate its effectiveness. This workflow enables collaborative behaviors among LLMs such as step-wise reflection and adjustment (*e.g.*, the third subfigure from the left of Figure 5), which is essential for long-sequence planning. (3) Our Spatial Geometric Computation Library facilitates language-driven manipulation of the physical world. As illustrated in the construction process in Figure 10, these processes involve diverse actions including attachment, removal, rotation, shifting, and connection, which collectively meet the action requirements of construction tasks. (4) The simulator provides environmental support for the evaluation phase. For instance, it can

place loads on bridges to test their load-bearing capacity and offer trajectory tracking for the entire launch process of rockets. Overall, these key components of **BuildArena**, including task design, library, and simulator, collectively provide robust support for evaluation, enabling it to function as an effective and reliable benchmark.

For ablation study on the agentic workflow, please refer to Table 6 of Appendix D. For more results about the decoding sensitivity, please refer to Table 7 of Appendix D. **BuildArena** can also use feedback from the simulator for closed-loop improvement. On **Support** Lv.1, closed-loop feedback brings failed samples to 100% success within five rounds (Table 8). On the more challenging **Lift** Lv.2, we further introduce an analyst agent that reviews simulator trajectories and failure causes to guide subsequent attempts; all three tested models achieve non-zero success from an initial 0% baseline (Table 9). Details are in Appendix D. We also analyze the impact of prompting protocol (zero-shot vs. one-shot) in Section 3.2.3.

To broaden the benchmark's coverage, we additionally evaluate three open-weight models (Qwen3.5-9B, Qwen3.5-27B, Ministral-14B). As shown in Table 4 of Appendix D, these models achieve non-trivial performance on several tasks (*e.g.*, Qwen3.5-27B reaches 35.9% success rate on **Support** Lv.1), while remaining substantially below the top proprietary models on higher difficulty levels, confirming that **BuildArena** is accessible to open-weight models while maintaining discriminative power.

We further note that the relative simplicity of structures observed in the main experiments is driven by task formulation and model design choices, rather than system limitations. When prompts explicitly emphasize structural complexity and visual elaborateness (see Appendix J.2 for the prompt), LLMs produce substantially more intricate constructions using the same module set (see Figure 11). This confirms that the benchmark framework itself supports complex constructions and is extensible to more demanding task specifications.

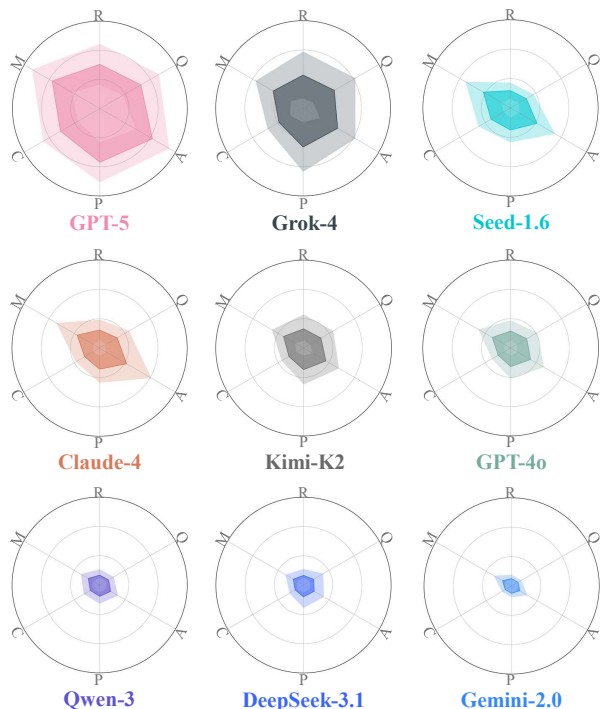

*Figure 7.* Performance of different LLMs against six dimensions of task difficulty: Quantification (Q), Robustness (R), Magnitude (M), Compositionality (C), Precision (P), Ambiguity (A).

## 3.2. Performance of LLMs

### 3.2.1. LIMITED PERFORMANCE OF CURRENT LLMS

From a complementary perspective, leveraging our **BuildArena** framework, current LLMs demonstrate elementary construction capabilities, as shown in Table 2. These capabilities are reflected in three aspects as follows. (1) In the **Transport** tasks, as difficulty increases from Lv.2 to Lv.3, all models adapt by scaling up the number of components to meet the increased payload size, thereby maintaining moving stability during simulation. Such patterns indicate that current LLMs effectively address challenges related to magnitude and ambiguity. (2) When explicit constraints are relaxed, LLMs attempt unconventional solutions: they propose propulsion-powered vehicles for **Transport** tasks and wheel-integrated bridge structures for **Support** tasks. In the latter case, LLMs explicitly state the need to utilize the automatic braking function of wheels to stabilize the bridge (see Figure 10, **Support** Lv.1 by DeepSeek-3.1). These behaviors highlight the potential of LLMs for creative exploration. (3) Notably, LLMs construct structures that align with real-world engineering practices, such as widely used steel trusses in bridges and differential steering systems in vehicles (see Figure 10, **Support** Lv.2 by Grok-4). This observation suggests that structural concepts learned from text are not purely symbolic but carry implicit spatial information, enabling LLMs to instantiate them as feasible 3D structures.

Beyond the construction process, we observe that the resulting structures in Figure 9 exhibit patterns consistent with established engineering design paradigms, despite the absence of explicit construction examples in the prompts. For instance, in the **Support** tasks, several models independently produce truss-like reinforcement configurations resembling steel girder bridges; in the **Transport** tasks, models adopt multi-wheeled chassis layouts with load-bearing platforms. These outcomes suggest that LLMs may carry implicit engineering knowledge acquired during pre-training, which can be activated and grounded into physically plausible 3D assemblies through our benchmark. This finding indicates that **BuildArena** has the potential to serve as a systematic testbed for probing and characterizing such latent engineering reasoning capabilities in LLMs.

However, these models still suffer from significant limitations. (1) In hierarchical assembly tasks, such as the **Support** task, LLMs' success rates drop sharply as the assembly complexity, *i.e.*, the number of bridge substructures, increases. This indicates that the models ability to cope with compositional constructions is generally weak. (2) In high-precision tasks with low robustness, such as the **Lift** tasks, the model's success rate is generally extremely low. As the difficulty increases, most success rates drop to zero. This shows that existing models, with the exception of GPT-5 and Grok-4, are unable to effectively accomplish tasks that are highly sensitive to accuracy.

Figure 6 shows the occupation of different failure reasons during the construction process. Several features can be extracted from it. (1) **Overlap Conflict and Face Occupied are the most difficult mistakes to avoid.** It indicates that LLM agents frequently fail to capture the updated spatial structures and make accurate next moves. (2) **Failure modes differ across task categories.** In the **Support** tasks, excess connection errors become more common, showing increasing attempts to reinforce the attachment. And in the **Lift** tasks, more misjudge of face status emerges since the structures have less modules and thus less redundant faces for attachment.

### 3.2.2. COMPARISON AMONG DIFFERENT LLMS

The performance of different LLMs across six difficulty dimensions is presented in Figure 7. It calculates the weighted score of each LLM across all the dimensions based on its ranking in each task, followed by averaging the scores across the nine task-difficulty combinations. Key observations are as follows. (1) GPT-5 and Grok-4 show competitive performance, which aligns with existing research (Phan et al., 2025; ARC Prize Foundation, 2025). (2) Aside from GPT-5 and Grok-4, different models show a high degree of similarity in the distribution of their capabilities to face varying difficulty. For each model, its strengths are

*Table 3.* Average ($n = 64$) performance on **Support** Lv.1 under zero-shot and one-shot prompting. The one-shot example achieves a maximum load of 1000.

| Model | Prompt | Success Rate (%)↑ | Indicator↑ |
|---|---|---|---|
| GPT-4o | Zero-shot | 40.6 | 181.2 |
| | One-shot | 25.0 (↓15.6) | 160.6 (↓20.6) |
| GPT-5 | Zero-shot | 85.9 | 324.9 |
| | One-shot | 65.6 (↓20.3) | 294.0 (↓30.9) |

consistently stretched in the Magnitude and Ambiguity dimensions, which is consistent with their performance in the **Transport** task in Table 2. In contrast, all LLMs exhibit consistent weaknesses across the other four dimensions. These findings provide clear directions for future improvements of LLMs.

### 3.2.3. ZERO-SHOT VS. ONE-SHOT PROMPTING

A natural question is whether few-shot prompting can improve construction performance. We compare zero-shot and one-shot prompting on the **Support** Lv.1 task, where the one-shot example is a strong construction achieving a maximum load of 1000 (see Appendix J.1 for the full prompt). As shown in Table 3, providing an example consistently degrades both success rate and indicator for both models. Because our tasks are open-ended reasoning problems, the desired structure must be derived through spatial reasoning and multi-step planning rather than imitation of a reference solution. Supplying a strong example reduces output diversity and biases models toward replicating the demonstrated design instead of generating novel, task-adapted structures. This finding justifies our adoption of zero-shot evaluation as the default benchmarking protocol.

### 3.2.4. COST ANALYSIS

Figure 8 presents the relationship between cost and performance. Detailed results are provided in Table 5. From these results, we observe that **massive inference does not guarantee high performance.** In all the three task categories, best construction results often consume only moderate numbers of tokens, whereas many failed attempts incur massive token usage. This shows that beyond a certain capability threshold, additional inference cost does not translate into better performance.

## 4. Conclusion and Limitations

In this work, we have introduced **BuildArena**, the first physics-aligned interactive benchmark designed to evaluate LLMs in construction-stage spatial assembly under physics constraints. While our work represents a first step toward the promising domain of LLM-based engineering construction,

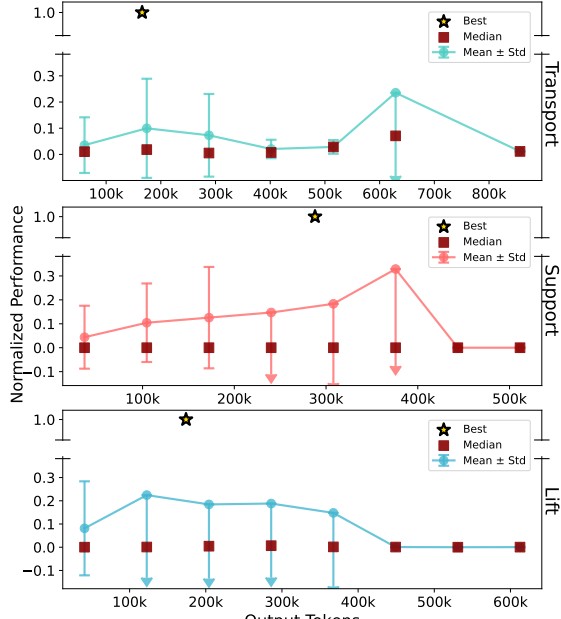

*Figure 8.* Trade-off between normalized performance and output token cost. Within each task family, the performance of the best single run is set to 1.0 and all other runs are normalized relative to it. Individual runs are grouped into bins by total output tokens consumed in one construction instance, and the mean/median statistics are computed within each bin. Longer output does not imply better results.

the following limitations remain. First, **BuildArena** evaluates spatial reasoning and multi-step planning within the Besiege simulation environment, which, despite its realistic physics, differs from real-world robotics and manufacturing constraints. We view this as a controlled testbed rather than a direct proxy for real manufacturing pipelines. Second, our main experiments adopt a single-shot setting. We provide preliminary closed-loop case studies (Tables 8 and 9 in Appendix D) showing that post-failure reflection can yield performance gains, but robust iterative refinement remains an open challenge. Third, the current task suite, while compact and discriminative, covers three task categories, and the limited diversity of basic units in the module library constrains the range of constructible objects.

## Acknowledgments

We are grateful to Spiderling Studios for creating Besiege, the inspiring physics sandbox that underpins our work. We also thank the developers of the open-source projects Lua Scripting Mod and Besiege Creation Import Addon for Blender for their valuable contributions to the community.

We also gratefully acknowledge the support of Westlake University Research Center for Industries of the Future. The content is solely the responsibility of the authors and does not necessarily represent the official views of the funding entities.

## Impact Statement

This paper advances the evaluation of LLMs in interactive, physics-constrained engineering construction, helping bridge language intelligence with the physical world. By systematically exposing where current frontier models fall short, this paper aims to provide the community a shared, reproducible testbed that motivates progress toward more physically grounded, reliable, and safe language model systems and training paradigms.

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

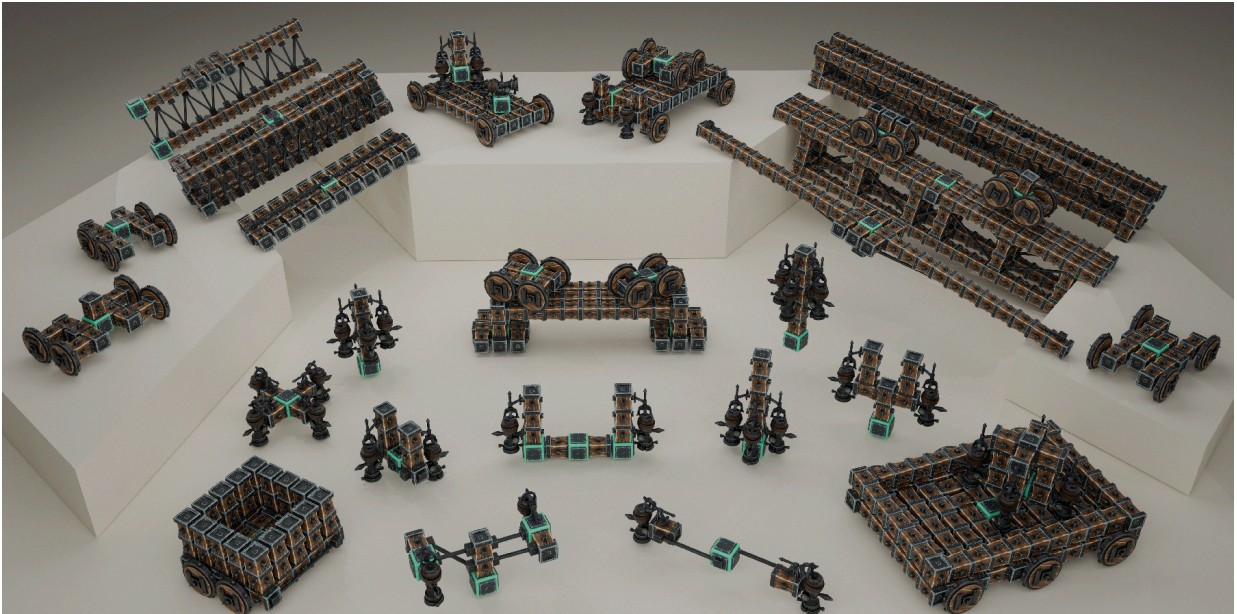

*Figure 9.* More examples of LLM construction results of **BuildArena**, spanning three tasks: **Transport** (vehicles), **Support** (bridges) and **Lift** (rockets with nozzles).

## A. Related Work

**LLM Capability Benchmarks.** There have been numerous research evaluating the capabilities of LLMs recently, but few focus on construction tasks as **BuildArena**. In long-term planning, popular benchmarks such as PlanBench (Valmeekam et al., 2023), PlanGenLLMs (Wei et al., 2025), and PlanningArena (Zheng et al., 2025) all operate in abstract and static settings, ignoring physical constraints. Physical reasoning benchmarks like PhYRE (Bakhtin et al., 2019), PIQA (Bisk et al., 2020), Newton (Wang et al., 2023), and ABench-Physics (Zhang et al., 2025c) do not involve assembly. Spatial understanding work, such as GeoGramBench (Luo et al., 2026), FloorplanQA (Rodionov et al., 2025), reveals LLM blind spots in real-world layout reasoning but omits construction. None of them tie LLM capabilities to physics-constrained engineering assembly, whereas **BuildArena** uses a physics sandbox to evaluate interactive construction.

**Physics Simulation Environments.** Advanced engines (MuJoCo (Todorov et al., 2012), Isaac Gym (Makoviychuk et al.)) and platforms (Autodesk (Autodesk Inc., 2024), SimScale (SimScale GmbH, 2024)) offer robust physics modeling, while digital twins create virtual structure representations (Dai et al., 2024). However, they lack integration with language interfaces for interactive construction, limiting their potential for following human instructions. LLM-based simulation studies focus on decision-making, not mechanical assembly (Kleiman et al., 2025; Gao et al., 2024), and robot-centric environments (Gazebo, DART-LLM (Wang et al., 2024b)) mainly target manipulation rather than structural building. **BuildArena** uniquely integrates a physics-aligned sandbox with a standardized language interaction protocol, tailored to evaluate the engineering construction driven by LLMs.

**AI-Driven Construction Automation.** AI applications in construction focus on parameterized design (Newton, 2019) and construction planning optimization (Zhang & Yang, 2025). Integrations with LLMs are still in the early stages (Ma12 et al.). A closely related concurrent study, BesiegeField (Zhang et al., 2025b), investigates machine design by casting construction as code generation tasks in Besiege (Spiderling, 2018). Our work differs from it in that BesiegeField primarily represents machines through a structured output specifying component relations, whereas we leverage the Spatial Computation Library (see Section 2.2) to continuously verify the LLM's actions, eliminating spatial conflicts in the final structures. BesiegeField demonstrates the effectiveness of RL finetuning for improving LLM performance on the construction tasks. However, their experiments indicate that current LLMs can not avoid spatial conflicts in the generated structures. The critical challenge of translating natural language to physically feasible structures remains unsolved.

## B. More Examples of Construction Results

More construction results are presented in Figure 9.

## C. More Examples of the Construction Process

More examples of construction process are presented in Figure 10.

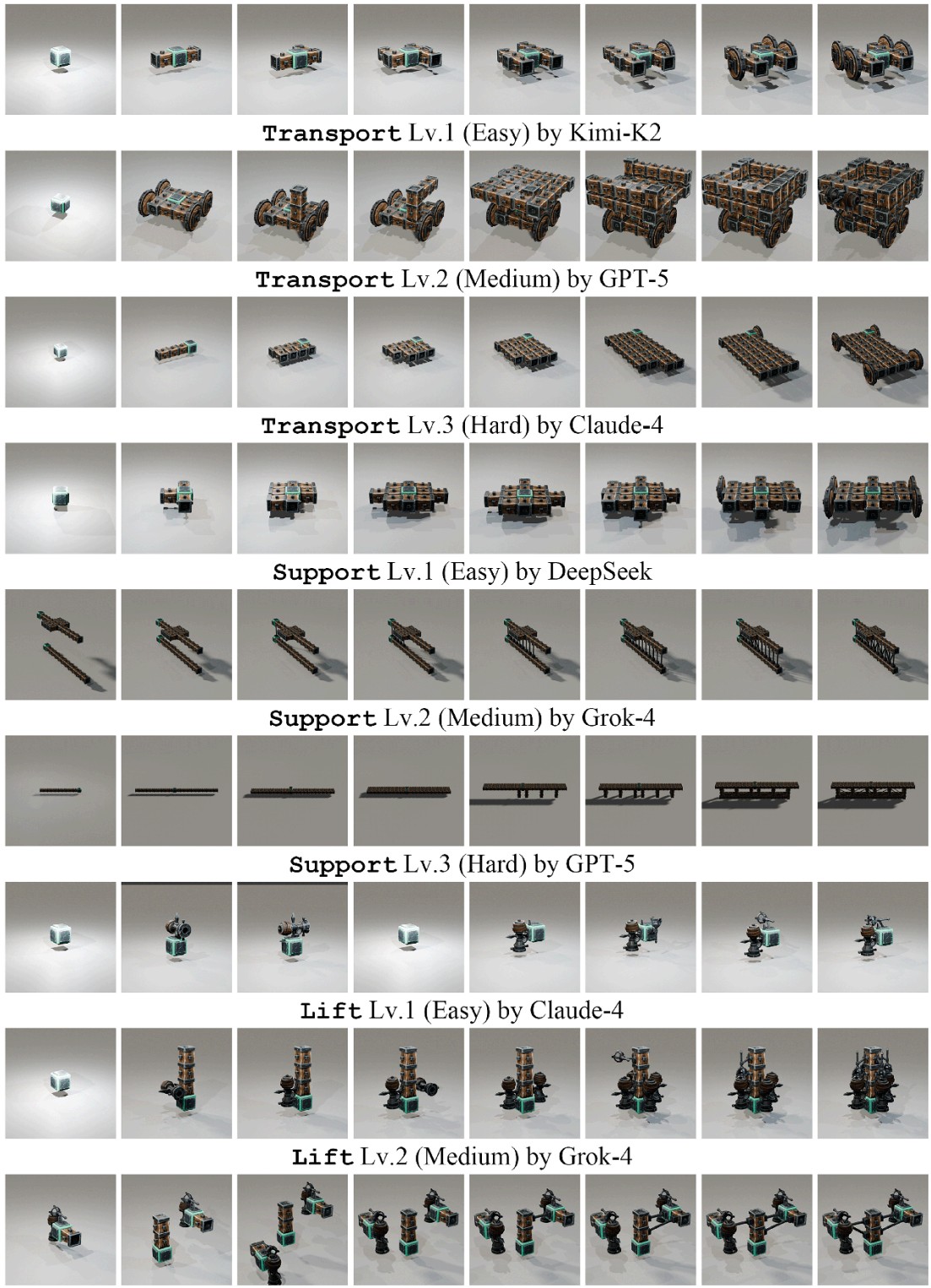

*Figure 10.* Examples of construction process across 9 tasks.

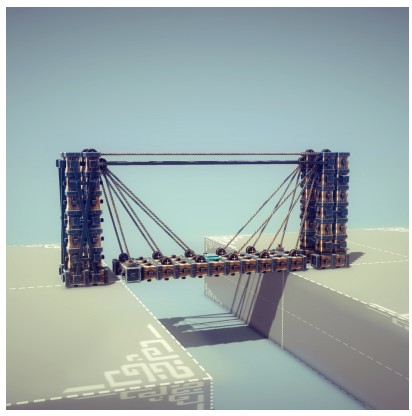 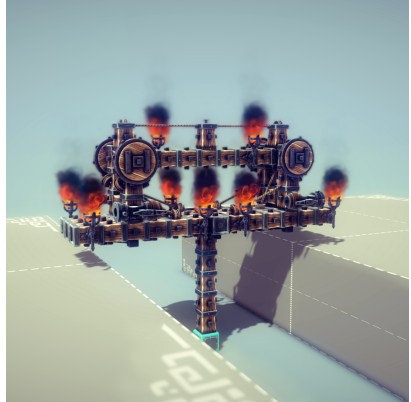 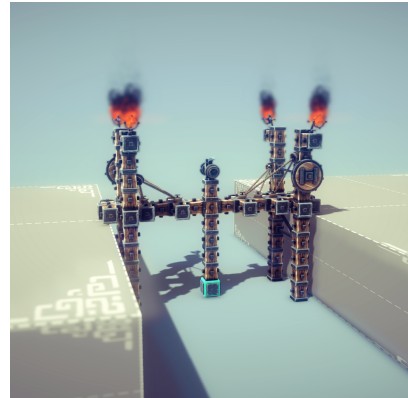

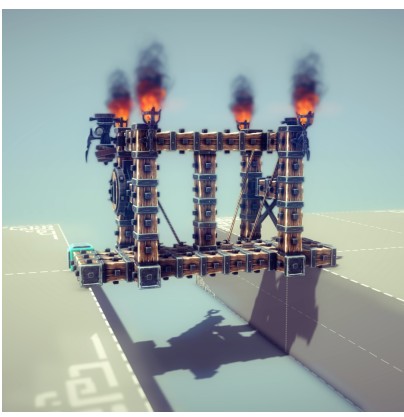 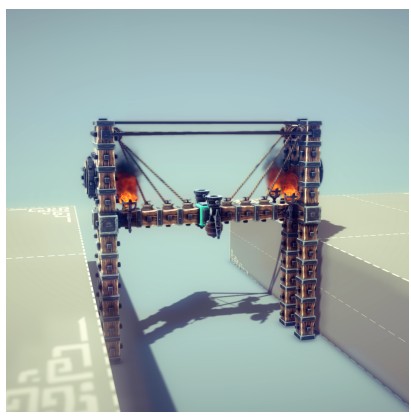 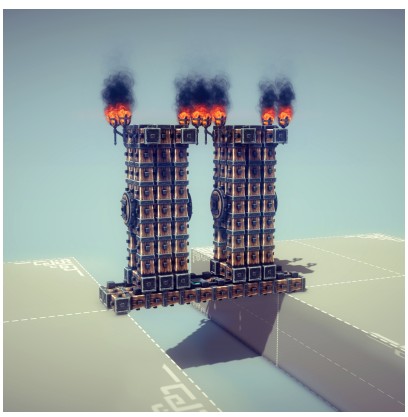

*Figure 11.* Complex bridge constructions produced by LLMs when explicitly prompted to emphasize structural complexity and visual elaborateness. These results demonstrate that the relative simplicity of structures in the main benchmark tasks is driven by task formulation rather than system limitations. The benchmark framework itself supports complex constructions using the same module set.

## D. More Experiment Results

**Open-weight Model Evaluation:** To broaden the benchmark's coverage beyond proprietary models, we evaluate three open-weight models: Qwen3.5-9B, Qwen3.5-27B, and Ministral-14B. As shown in Table 4, Qwen3.5-27B achieves 35.9% success rate on **Support** Lv.1 and 12.5% on **Transport** Lv.1/Lv.3, while their overall performance remains substantially below the top proprietary models (GPT-5, Grok-4), particularly on higher difficulty levels where most success rates drop to zero. These results confirm that **BuildArena** is accessible to open-weight models while maintaining discriminative power.

**Cost Details of Models:** Detailed cost statistics are presented in the Table 5.

**Multi-Agent Pipeline Ablation:** We compare the five-role workflow against simpler controller variants on the **Support** Lv.1 task using the Seed-1.6 model. As shown in Table 6, the full five-role workflow (Planner, Drafter, Reviewer, Guidance, Builder) outperforms simplified alternatives on this task. The single-agent Builder baseline suffers from an extremely high invalid action rate of 44.7%, indicating difficulty in generating valid construction sequences without structured guidance. We note that on more challenging Lv.2 tasks, the advantage of the multi-agent workflow is less consistent and depends on the specific task and model combination. We therefore treat the current workflow as a fixed, shared evaluation protocol so that all models are compared under the same agentic setting, rather than prescribing it as the universally optimal pipeline design. Under this unified protocol, the benchmark exhibits meaningful discriminative power across models.

**Zero-shot vs. One-shot Prompting:** We adopt zero-shot evaluation as the default protocol for **BuildArena**. As shown in Table 3 (Section 3.2.3), providing a strong one-shot example (achieving a maximum load of 1000) on the **Support** Lv.1 task consistently degrades both success rate and indicator for GPT-4o and GPT-5. This confirms that these open-ended construction tasks require genuine spatial reasoning rather than pattern imitation, supporting the zero-shot protocol as a fairer measure of intrinsic construction capability.

*Table 4.* Average ($n = 64$) performance of open-weight models on different tasks across task levels Lv.1 (easy), Lv.2 (medium), and Lv.3 (hard). The indicator means maximum displacement for **Transport**; maximum load for **Support**; TWR for Lv.1 and maximum height for Lv.2, Lv.3 of **Lift**.

| Task | Model | Number of Parts | | | Success Rate (%)↑ | | | Indicator↑ | | |
|------|-------|------|------|------|------|------|------|------|------|------|
| | | Lv.1 | Lv.2 | Lv.3 | Lv.1 | Lv.2 | Lv.3 | Lv.1 | Lv.2 | Lv.3 |
| **Transport** | Qwen3.5-9B | 3.5 | 4.3 | 2.4 | 0.0 | 0.0 | 0.0 | 1.5 | 0.8 | 0.5 |
| | Qwen3.5-27B | 7.0 | 18.3 | 48.0 | 12.5 | 3.1 | 12.5 | 14.5 | 5.1 | 8.4 |
| | Ministral-14B | 10.1 | 10.1 | 14.2 | 4.7 | 0.0 | 1.6 | 8.1 | 1.9 | 2.4 |
| **Support** | Qwen3.5-9B | 5.4 | 0.3 | 0.0 | 3.1 | 0.0 | 0.0 | 7.8 | 0.0 | 0.0 |
| | Qwen3.5-27B | 23.6 | 29.1 | 61.0 | 35.9 | 10.9 | 0.0 | 166.0 | 36.1 | 0.0 |
| | Ministral-14B | 18.6 | 2.1 | 1.2 | 1.6 | 0.0 | 0.0 | 3.5 | 0.0 | 0.0 |
| **Lift** | Qwen3.5-9B | 3.0 | 3.0 | 0.2 | 1.6 | 0.0 | 0.0 | 0.4 | 1.7 | 0.1 |
| | Qwen3.5-27B | 4.5 | 5.0 | 0.5 | 10.9 | 0.0 | 0.0 | 0.9 | 1.8 | 0.5 |
| | Ministral-14B | 5.9 | 11.4 | 0.5 | 12.5 | 0.0 | 0.0 | 0.8 | 2.7 | 0.1 |

**Decoding Sensitivity:** To assess the robustness of our pipeline to decoding configurations, we conduct ablation experiments varying temperature settings on the **Support** Lv.1 task using the Seed-1.6 model. As shown in Table 7, our default configuration ($temperature = 0.5$, top-p $= 0.7$) achieves the highest success rate of 45.3%. Varying temperature shows modest impact: $temperature = 0.0$ yields 43.8% success rate with slightly higher invalid actions (2.3%), while $temperature = 1.0$ achieves the highest maximum load indicator (199.1) but lower success rate (42.2%). Overall, performance variations across configurations remain within a reasonable range, with invalid action rates consistently low, demonstrating that the benchmark remains discriminative and solvable across different decoding settings.

**Closed-loop Feedback Refinement:** To evaluate the potential of iterative refinement through closed-loop feedback, we conducted additional experiments on the **Support** Lv.1 task. We randomly selected 18 failed samples from the Seed-1.6 model and incorporated simulation results as feedback for subsequent refinement turns. As shown in Table 8, closed-loop feedback substantially improves construction performance: 72% (13/18) of samples pass after the first refinement turn, increasing to 83% (15/18) after Turn 3, and reaching a 100% (18/18) success rate by Turn 5. It demonstrates that integrating simulation feedback enables models to iteratively correct design flaws and meet task requirements. However, the performance gain comes at the cost of increased computational expense due to multiplied LLM inference rounds. While closed-loop refinement is valuable for practical engineering applications, we adopt single-turn evaluation as the default benchmark setting to better distinguish the inherent construction capabilities across different LLMs.

**Closed-loop with Analyst Agent:** To further explore post-execution reflection, we conduct an additional closed-loop experiment on the **Lift** Lv.2 task using three models with 0% single-shot success rate. We introduce an analyst agent with access to code and file tools, which reviews the machine structure, simulator feedback, and trajectories, summarizes failure causes, and appends improvement suggestions to the next-round prompt. As shown in Table 9, the gains are modest but meaningful: all three models achieve non-zero success rates in later rounds. DeepSeek-3.1 reaches 3.1% success rate at Round 4, while Qwen-3 achieves 1.6% at Rounds 3 and 5. These results suggest that post-failure reflection is feasible and can sometimes rescue unsuccessful construction policies, although robust reflective learning remains an open challenge.

# E. Definitions of Failure Reasons

- **Violate Module Restriction:** Attempting to use a module that is not provided in (or does not exist in) the module library.

- **Misjudge Block Presence:** Attempting to edit a block that does not exist in the current structure.

- **Connector Misuse:** Using the connector module incorrectly; a connector can only connect two *already-existing* attachable faces and cannot be directly added into the structure as a standalone module.

- **Face Occupied:** Attempting to add a new module onto an attachable face that is already occupied (*i.e.*, already connected/attached).

*Table 5.* Average ($n = 64$) cost comparison of models on different tasks across levels Lv.1 (easy), Lv.2 (medium), and Lv.3 (hard). The number of input/output tokens (# Input/Output Tokens) represents the cumulative total across multiple LLM requests required to complete one task instance on average.

| Task | Model | # Input Tokens ($\times 10^3$)↓ | | | # Output Tokens ($\times 10^3$)↓ | | | # LLM Requests↓ | | |
|---|---|---|---|---|---|---|---|---|---|---|
| | | Lv.1 | Lv.2 | Lv.3 | Lv.1 | Lv.2 | Lv.3 | Lv.1 | Lv.2 | Lv.3 |
| **Transport** | GPT-5 | 350.6 | 3306.0 | 5630.6 | 95.4 | 305.4 | 302.8 | 54.9 | 169.6 | 220.2 |
| | GPT-4o | 326.7 | 546.2 | 458.1 | 22.1 | 11.1 | 11.6 | 63.5 | 69.4 | 74.7 |
| | Claude-4 | 203.0 | 264.9 | 751.5 | 18.5 | 19.1 | 25.3 | 41.3 | 48.0 | 74.3 |
| | Grok-4 | 233.5 | 368.3 | 1259.9 | **8.9** | **10.8** | 15.5 | 40.7 | 49.3 | 98.4 |
| | Gemini-2.0 | 382.5 | 371.6 | **361.7** | 11.0 | **10.8** | **9.3** | 65.4 | 63.1 | **63.9** |
| | DeepSeek-3.1 | 252.8 | 469.4 | 715.5 | 18.5 | 23.6 | 28.2 | 49.8 | 67.9 | 80.1 |
| | Qwen-3 | 473.8 | 381.1 | 841.3 | 18.5 | 19.9 | 21.8 | 59.7 | 51.1 | 72.2 |
| | Kimi-K2 | 635.5 | 1099.6 | 968.7 | 13.9 | 16.0 | 14.2 | 82.2 | 96.8 | 99.8 |
| | Seed-1.6 | **197.1** | **248.7** | 899.3 | 41.0 | 43.2 | 63.7 | **34.1** | **41.1** | 81.5 |
| **Support** | GPT-5 | 1063.6 | 4040.2 | 7348.7 | 99.8 | 164.5 | 209.9 | 89.9 | 179.7 | 242.0 |
| | GPT-4o | 1008.6 | 748.7 | 1671.8 | 19.5 | 19.0 | 22.5 | 102.5 | 135.7 | 192.5 |
| | Claude-4 | **116.3** | 548.5 | 1134.4 | 8.6 | 30.9 | 40.3 | **26.7** | 94.4 | 135.1 |
| | Grok-4 | 301.8 | 545.1 | 1152.0 | **7.6** | **11.9** | 14.7 | 45.5 | 69.4 | 91.1 |
| | Gemini-2.0 | 425.3 | 1413.9 | 2308.0 | 10.4 | 20.1 | 22.2 | 62.7 | 142.2 | 186.5 |
| | DeepSeek-3.1 | 484.4 | **299.0** | **424.8** | 22.1 | 19.3 | 18.4 | 70.5 | **57.0** | 64.8 |
| | Qwen-3 | 880.1 | 817.5 | 1263.5 | 17.7 | 23.9 | 23.3 | 63.8 | 89.4 | 104.0 |
| | Kimi-K2 | 508.8 | 1750.3 | 861.1 | 8.6 | 22.6 | **7.3** | 60.9 | 160.0 | **64.2** |
| | Seed-1.6 | 880.3 | 2043.2 | 5423.0 | 51.5 | 112.2 | 165.4 | 78.3 | 165.5 | 293.2 |
| **Lift** | GPT-5 | 197.8 | 350.8 | 895.4 | 124.0 | 138.6 | 276.8 | 31.9 | 48.5 | 101.0 |
| | GPT-4o | 345.7 | 232.3 | 423.6 | 23.7 | 8.7 | 13.8 | 51.2 | 50.8 | 88.1 |
| | Claude-4 | 279.8 | 376.0 | 386.8 | 22.6 | 28.8 | 30.6 | 44.9 | 50.7 | 68.9 |
| | Grok-4 | **103.4** | **180.9** | **128.0** | 6.7 | **8.4** | **10.2** | **24.2** | **33.0** | **34.3** |
| | Gemini-2.0 | 290.5 | 266.1 | 445.1 | 7.1 | 10.3 | 14.1 | 40.4 | 47.6 | 80.3 |
| | DeepSeek-3.1 | 317.3 | 401.6 | 396.8 | 22.5 | 24.7 | 29.5 | 54.4 | 61.3 | 76.0 |
| | Qwen-3 | 483.7 | 987.0 | 877.2 | 22.0 | 28.6 | 18.1 | 49.7 | 76.5 | 76.5 |
| | Kimi-K2 | 288.8 | 885.8 | 715.2 | 7.4 | 11.1 | 13.4 | 46.5 | 84.6 | 96.5 |
| | Seed-1.6 | 227.4 | 233.3 | 244.3 | 51.4 | 52.1 | 62.9 | 35.0 | 35.8 | 50.3 |

- **Excess Connection:** Attempting to use a connector to link two attachable faces that are already directly attached.

- **Misjudge Face Presence:** Attempting to perform a connect/add operation on a face that does not exist on the target module.

- **Multiple Initialization:** Re-initializing construction after the environment has already been initialized.

- **Initialization Absent:** Starting construction actions without performing initialization (thus failing to add the starting block into the environment).

- **Misjudge Spin Capability:** Attempting to change the default spin direction of a non-rotatable module.

- **Overlap Conflict:** An add/move/rotate operation causes physical volume overlap among modules, resulting in an overlap conflict.

# F. 3D Spatial Geometric Computation Library

## F.1. Module Space

The modules space $\mathcal{V}$ is a complete collection of basic module types like small wooden block, powered wheel, water cannon, torch, brace and winch that can be combined to build complex objects. The state in the construction procedures is represented

*Table 6.* Ablation study on multi-agent pipeline for Seed-1.6 model on `Support` Lv.1 task ($n = 64$ samples). The Indicator measures the maximum load the bridge can support. Metrics are averaged across samples. Performance changes relative to our workflow are shown in parentheses.

| Workflow | Number of Parts | Success Rate (%)↑ | Indicator (Max Load)↑ | Invalid-Action Rate (%)↓ |
|---|---|---|---|---|
| **Five-role Multi-agent** | **33.4** | **45.3** | **197.4** | **1.4** |
| Guidance-Builder | 20.2 | 4.7 ($-40.6$) | 31.4 ($-166.0$) | 1.7 ($+0.3$) |
| Builder | 19.8 | 22.6 ($-22.7$) | 75.5 ($-121.9$) | 44.7 ($+43.3$) |

*Table 7.* Ablation study on decoding strategies for Seed-1.6 model on `Support` Lv.1 task ($n = 64$ samples). Bold indicates our default configuration. Underline indicates the best performance for each metric.

| Temperature | Top-p | Number of Parts | Success Rate (%)↑ | Indicator (Max Load)↑ | Invalid-Action Rate (%)↓ |
|---|---|---|---|---|---|
| **0.5** | **0.7** | **33.4** | **45.3** | **197.4** | **1.4** |
| 0.0 | 0.7 | 30.0 | 43.8 | 168.8 | 2.3 |
| 1.0 | 0.7 | 29.8 | 42.2 | 199.1 | 1.8 |

*Table 8.* Closed-loop feedback refinement results on `Support` Lv.1 task ($n = 18$ failed samples). The Indicator measures the maximum load the bridge can support. Metrics are averaged across samples at each refinement turn.

| Metric | Round 1 | Round 2 | Round 3 | Round 4 | Round 5 |
|---|---|---|---|---|---|
| Success Rate (%)↑ | 0.0 | 72.2 | 83.3 | 94.4 | 100 |
| Indicator (Max Load)↑ | 0.0 | 422.6 | 477.2 | 568.8 | 586.9 |
| Number of Parts | 26.5 | 61.0 | 49.8 | 82.7 | 43.0 |
| Invalid-Action Rate (%)↓ | 1.2 | 1.1 | 2.2 | 0.9 | 1.2 |
| # LLM Requests↓ | 56.8 | 128.8 | 108.2 | 171.7 | 89.0 |

*Table 9.* Multi-turn closed-loop construction results on `Lift` Lv.2 task with analyst agent ($n = 64$ samples). Round 1 corresponds to single-shot construction without closed-loop feedback.

| Metric | Model | Round 1 | Round 2 | Round 3 | Round 4 | Round 5 |
|---|---|---|---|---|---|---|
| Success Rate (%)↑ | DeepSeek-3.1 | 0.0 | 0.0 | 1.6 | 3.1 | 0.0 |
| | Qwen-3 | 0.0 | 0.0 | 1.6 | 0.0 | 1.6 |
| | Seed-1.6 | 0.0 | 0.0 | 1.6 | 0.0 | 0.0 |
| Indicator (Max Height)↑ | DeepSeek-3.1 | 3.8 | 3.0 | 3.3 | 3.5 | 2.0 |
| | Qwen-3 | 3.1 | 2.5 | 3.0 | 1.9 | 2.7 |
| | Seed-1.6 | 2.6 | 1.9 | 2.6 | 1.5 | 1.4 |

as a triple $S = \langle V, \mathcal{P}, c \rangle$. Here, $V \subset \mathcal{V}$ denotes the set of modules involved in the current structure; The projection operator $\mathcal{P} : V \to \mathrm{SE}(3)$ maps each module in $V$ element-wisely to its 3D pose; and $c = \langle (t_1, k_1, \Delta t_1), \ldots, (t_n, k_m, \Delta t_n) \rangle$ forms a control sequence, where $t_n$ is the timestamp of pressing control key $k_m$, $\Delta t_n$ is the press duration, and overlapping key-press operations are permitted.

To be more specific, each module $v \in \mathcal{V}$ is characterized by four attributes, expressed as $v = \langle \mathcal{F}, G, \gamma, \pi \rangle$. Geometric attribute $G$ encodes mesh, collision shape, initial orientation, and relative coordinates/orientations of connectable faces; $\mathcal{F}$ is a finite set of connectable faces where with each face is associated with a normal vector derived from $\mathcal{P}(v)$ and $G$, described via natural-language-aligned terms like letter labels and angular coordinates; $\gamma$ represents physical/functional parameters such as mass, rotational speed, thrust; control attributes $\pi$ maps control keys to actions and spin directions. Additionally, each module is accompanied by a natural-language summary $c_{\mathrm{init}}(G, \gamma, \pi)$ for initial prompts, with few-shot examples for modules (e.g., Powered Wheel) to help LLM agents align with critical functional info, such as rolling/jetting directions.

## F.2. Action Space

The action space comprises five categories of operators that cover the whole construction process: `Build`, `Refine`, `Assemble`, `Control`, and `Query`. A sequence of these actions forms a trajectory $\bar{a} = \langle (a_1, r_1), (a_2, r_2), \ldots, (a_T, r_T) \rangle$, where each $a_i \in \mathcal{A}$ and $r_i$ denotes return information, including success / failure codes and natural-language descriptions of spatial structures. These categories are defined as follows:

`Build`: Enables core construction operations in the simulated environment, including module attachment, connection, removal, resetting, rotation, translation, and reversal.

`Refine`: When the structural state generated in the Build phase contains rotating modules, a subsequent Refine phase follows to allow fine-tuning of the structural state, in an attempt to ensure that the rotating modules have a reasonable rotation direction.

`Assemble`: If multiple substructures were built, an Assembly phase is then initialized, allowing the reuse of former construction results as building components.

`Control`: Manages control-related functionalities, such as updating action-to-control-key mappings in $\pi_v$ and appending control operations $(t, k, \Delta t)$ to the control sequence $\delta$ within $s$.

`Query`: Retrieves natural-language descriptions of structural states, including: a summary of the overall state $s$ as $r_t = c_s(\mathcal{V}, \mathcal{P})$; detailed module $v$ information as $r_t = c_v(\mathcal{F}_v, p_v)$; function-to-key mappings for control-enabled modules in $s$ as $r_t = c_s(\pi_1, \pi_2, \ldots, \pi_v)$; descriptions of $s$'s control sequence as $r_t = c_s(\delta)$; and function-to-pose mappings for control-enabled modules in $s$ as $r_t = c_s(p_1, p_2, \ldots, p_v)$.

## F.3. Besiege as Simulation Backend

Besiege is a physics-based construction sandbox game environment that enables the assembly and simulation of mechanical structures using modular components. It features a realistic physics engine, validated through extensive community use, which aligns closely with real-world physical principles. The environment includes a diverse set of structural and functional modules (over 70 types), allowing for the iterative construction of complex objects such as vehicles and static supports. These can be tested in simulated scenarios, with support for multiplayer validation and access to over 200,000 user-generated designs via an integrated workshop. Besiege simulates realistic dynamics, including gravity, friction, and module interactions, using Unity's physics system. Our library interfaces indirectly with Besiege by replicating its core operations (e.g., module attachment and control sequencing) without direct API access. In this work, we leverage Besiege as a neutral platform for evaluating language-driven construction under physical constraints, emphasizing its modular building system and simulation fidelity.

## F.4. Library API

The library implements the core functionalities for managing 3D spatial operations in the **BuildArena** framework. It handles state updates, geometric transformations, and constraint validations, bridging LLM-generated instructions to Besiege's physics simulation. Functions are organized into tool groups for modular use: **control** for sequencing actions, **build** for assembling structures, **refine** for post-attachment adjustments, **default** for querying states, and **build_only** for initialization. Below, we highlight representative functions from each group.

### F.4.1. CONTROL TOOL GROUP

This group manages timed control sequences for powered modules, enabling dynamic behaviors in simulated environments.

*add_control_sequence*: Facilitates the addition of timed inputs to simulate machine operations, essential for tasks requiring sequential activation.

```python
def add_control_sequence(time: float, key: str, hold_for: float) -> str:
    """Add a new control sequence entry."""
    # Implementation: Append to sequence list with validation
    return "Sequence added successfully."
```

*review_control_config*: Provides visibility into current control mappings, supporting iterative debugging during construction.

```python
def review_control_config() -> str:
    """A tool to review the current control configuration."""
    # Implementation: Aggregate and format control data
    return "Control config: [list of keys and actions]"
```

### F.4.2. BUILD TOOL GROUP

This group supports the core assembly of structures, including attachments and connections under geometric constraints.

*attach_block_to*: Enables precise module placement on existing structures, enforcing face-based alignment for stable builds.

```python
def attach_block_to(base_block: Union[str, int], face: str, new_block: str, note: str) ->
      str:
    """Attach a new block to a face of an existing block."""
    # Implementation: Compute pose, check collisions, update state
    return "Block attached successfully."
```

*connect_blocks*: Establishes reinforced links between modules, crucial for enhancing structural integrity in complex designs.

```python
def connect_blocks(block_a: Union[str, int], face_a: str, block_b: Union[str, int], face_b
      : str, connector: str, note: str) -> str:
    """Connect two blocks using a connector."""
    # Implementation: Validate distance, add connector module
    return "Blocks connected successfully."
```

### F.4.3. REFINE TOOL GROUP

This group allows fine-tuning of module positions and orientations after initial placement, aiding in overlap resolution.

*twist_block*: Adjusts rotational alignment to optimize functional orientations, particularly for directional components.

```python
def twist_block(block_id: Union[str, int], angle: float) -> str:
    """Twist a block clockwise relative to its rooted surface."""
    # Implementation: Apply rotation matrix, update pose
    return "Block twisted successfully."
```

### F.4.4. DEFAULT TOOL GROUP

This group provides state inspection tools, ensuring accurate feedback for LLM reasoning loops.

*get_machine_summary*: Offers a high-level overview of the current build state, mandatory for final validations before simulation.

```python
def get_machine_summary() -> str:
    """Get the latest state of the machine without face captions."""
    # Implementation: Summarize blocks and poses
    return "Machine summary: [overview details]"
```

F.4.5. BUILD-ONLY TOOL GROUP

This group handles initialization, setting the foundation for new constructions.

*start*: Initializes the build environment with a starting module, incorporating initial offsets for custom positioning.

```python
def start(init_shift: List[float], init_rotation: List[float], note: str) -> str:
    """Start to build the machine by creating and positioning the starting block."""
    # Implementation: Create initial block, apply transformations
    return "Starting block positioned."
```

### F.5. Library Fidelity Validation

Since the geometric computations in Besiege are closed-source, a natural concern is whether our library faithfully reproduces the simulator's internal construction logic. To validate fidelity, we manually reconstructed a 49-block machine in Besiege following the exact same LLM-generated action sequence, saved the result, and compared it with the structure produced by our library at the block level.

As shown in Figure 12, the discrepancies are extremely small across all 49 build steps. For position error (L2 norm), the maximum is $1.41 \times 10^{-6}$ unit length and the mean is $9.14 \times 10^{-7}$ unit length. For orientation error, the maximum is $2.43 \times 10^{-5}$ degrees and the mean is $1.38 \times 10^{-5}$ degrees. These errors are negligible relative to the scale of construction modules (which have dimensions on the order of 1 unit) and far below any threshold that could affect simulation outcomes. We therefore did not observe evidence that library inaccuracies cause false negatives in evaluation.

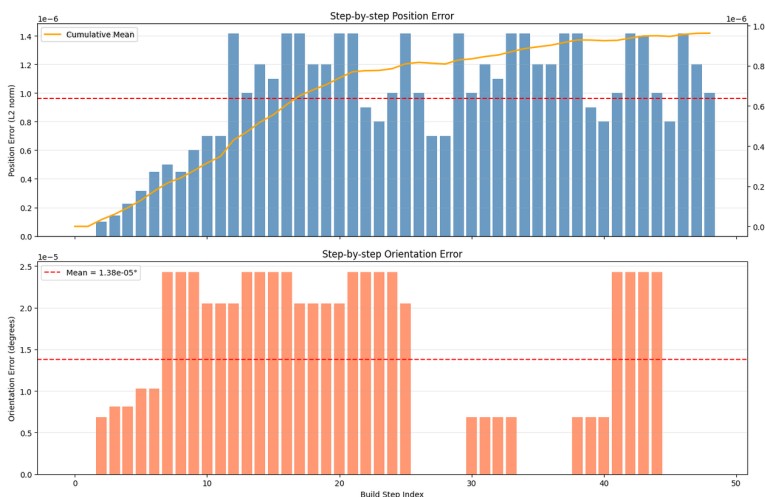

*Figure 12.* Step-by-step fidelity comparison between our library and the Besiege simulator on a 49-block machine. **Top:** Position error (L2 norm) per build step, with cumulative mean shown in orange. The mean position error is $9.14 \times 10^{-7}$ unit length. **Bottom:** Orientation error per build step. The mean orientation error is $1.38 \times 10^{-5}$ degrees.

## G. Experiments Details

### G.1. Model Snapshots

- Grok-4: grok4-0709
- GPT-4o: gpt-4o
- GPT-5: gpt-5-2025-08-07
- Claude-4: claude-sonnet-4-20250514
- Gemini-2.0: gemini-2.0-flash
- DeepSeek-3.1: deepseek-chat (DeepSeek-V3.1)

- Seed-1.6: doubao-seed-1-6-250615

- Kimi-K2: kimi-k2-turbo-preview

- Qwen-3: qwen-plus (Qwen3 series)

- Qwen3.5-9B: qwen3.5-9b

- Qwen3.5-27B: qwen3.5-27b

- Ministral-14B: ministral-14b

All models are accessed via official API endpoints encapsulated by AutoGen (Wu et al., 2024) framework.

### G.2. Decoding Parameters

- Temperature: 0.5

- Top-p: 0.7

### G.3. Basic Modules

For our experiments, we have defined a set of six basic modules that serve as the fundamental building blocks for all tasks. This curated selection—comprising the small wooden block, powered wheel, water cannon, torch, brace, and winch—is sufficient to realize the necessary constructions without introducing overpowered or overly specialized parts. The detailed descriptions and specifications of each module are presented below.

#### G.3.1. 4 KINDS OF AVAILABLE BLOCKS

**Powered Wheel (shape: [2, 2, 0.5], mass: 1.0)**

```
Description: A powered wooden wheel (diameter = 2, thickness = 0.5) rotates at a constant
    speed of 100 rpm, and automatically brakes when the wheel stops.  Each powered wheel
    can be individually controlled to rotate forward or backward by pressing and holding
    configurable control keys. The wheel's motion is governed by the following constraints
    : - The wheel's rotation axis is perpendicular to the attached face. - The rolling
    direction is always parallel to the attached face. - For example, if the attached face
     is a horizontal face, the wheel will also be horizontal; if the attached face is a
    vertical face, the wheel will also be vertical. - For example, if the wheel is
    attached to a side face, the wheel will be rotating parallel to the side face and the
    rolling direction is perpendicular to the side face. - For example, if the wheel is
    attached to a bottom face, the wheel will be rotating parallel to the bottom face and
    unable to roll effectively.
```

**Small Wooden Block (shape: [1, 1, 1], mass: 0.3)**

```
Description: Small wooden cubic block with shape of [1, 1, 1]
```

**Torch (shape: [1.5, 0.5, 0.5], mass: 1)**

```
Description: The torch flame sets flammable blocks, structures, and entities on fire.  It
    can be extinguished by Water Cannons (and Steam Cannons!), and reignited by other
    burning blocks. Their most common use is in heating water cannons so they produce
    steam, particularly in vanilla builds  However, they can be extinguished by steam
    plumes, so care must be taken not to fly backwards.  The torch will generate a
    spherical heating area with a radius of 0.3 unit in front of its flame nozzle
    direction (that is, the position of the torch body plus the orientation vector).  All
    objects in this area will be heated or ignited by the flame. They can also be used for
     setting fire to things for destructive purposes. Torches have no attachable faces for
     further attachment or connection. The Torch is shaped as a short horizontal support (
    length of 0.5), and a vertical shaft (length of 1), the flame is at the end of the
    vertical shaft. For example, if the torch is attached to a vertical face (face center
```

```
     is [0.5, 0, 0]) and points upwards, the torch coordinates will be [1, 0, 0] since the
     horizontal support of the torch has a offset of 0.5 from the attached surface, and the
      heating area will be a sphere with radius 0.3 centered at [1, 0, 1] since the length
     of the vertical shaft is 1.
```

### Water Cannon (shape: [2, 2, 1], mass: 1.5)

```
Description: The Water Cannon sprays water in a fixed direction, which is determined by
     the attachment and orientation of the water cannon. Generates constant recoil force of
      1.6 units of mass at normal gravity. The recoil force is not affected by speed or
     external conditions. Each water cannon can be individually controlled to fire by
     pressing and holding a configurable control key. Water Cannon has no attachable faces
     for further attachment or connection. Steam Mode: If any part of the water cannon is
     heated, it will fire steam instead of water and deliver 8.6 times the regular recoil
     force. Water Cannon has a peanut-shaped body (narrower in the middle than at the two
     ends, inlet and outlet are at the two ends) with length of 1.75, width of 1, and
     height of 1. The middle part of the water cannon is narrower, making it hard to be
     directly heated if the heat source is small. For example, if the water cannon is
     attached to a vertical face (face center is [0.5, 0, 0]) and points downwards, the
     water cannon center coordinates will be [1, 0, 0] since the connection part of the
     water cannon has a offset of 0.5 from the surface, and the water cannon inlet will be
     at [1, 0, 0.75] and the water cannon outlet will be at [1, 0, -1] with a shape of 1.75
     x1x1 cylinder (narrower in the middle than at the two ends).
```

G.3.2. 2 KINDS OF AVAILABLE CONNECTORS:

### Brace (mass: 0.5)

```
Description: The Brace is a block that can be used to connect two separated blocks with a
     solid hinge. It can be used to enhance two blocks that are already connected together,
      or to assemble structures that are separated in the space. The mass of this block is
     always the same regardless of the length. Brace must be connected between two
     attachable faces of existing blocks, it cannot be directly attached to a single block.
```

### Winch (mass: 0.4)

```
Description: The Simple Rope + Winch (simply as Winch or Rope) is a machine block composed
     of two winches at its end node which connects two blocks by a variable-length rope.
     Winch must be connected between two attachable faces of existing blocks, it cannot be
     directly attached to a single block.
```

## G.4. Simulation Details

All simulations are conducted on Besiege v1.75 (build 23370) with Lua Scripting Mod (for controlling and logging), using the Steam distribution on Windows, performed in the native physics settings of the game and executed by a unified automation script. Motion trajectories are recorded at a sampling rate of 25 Hz for subsequent quantitative analysis.

# H. Task details

Prompts of all the three task categories are listed as follows.

## H.1. Transport

### H.1.1. EASY (Lv.1)

```
**Constraints:**
- Use only one sub-structure.
- The vehicle must have at least four wheels.
- The vehicle must be capable of forward driving and demonstrate a steering mechanism.
- Conventional steering mechanisms (e.g., rotating front wheels relative to the body) are
     not available with the provided blocks. Alternative steering strategies must be
     employed.
```

```
**Goal:**
- Drive the vehicle from the starting position (x=0, y=0) on the ground to the target
   position (x=10, y=10) on the ground (north-east direction) in the simulation
   environment.

**Evaluation Protocol:**
- The vehicle will be placed at (x=0, y=0) on the ground in the simulation environment.
- An open-loop control sequence will be programmed by a specialized AI agent following
   your plan, consisting of a list of commands with the format:
- [time: when to press the control key, command: the control key to press, duration: how
   long to hold the key]
- The trajectory of the vehicle will be recorded as feedback and optimized over three
   trials by adjusting the control sequence.
- The final score will be the best score across the three trials.

**Scoring Metrics:**
- *Trajectory Deviation:* Distance between the actual trajectory and the ideal straight-
   line path from start to target (smaller is better).
- *Structure Stability:* Whether the vehicle remains intact during driving (higher
   stability is better).
- *Time Efficiency:* Time taken to reach the target position (shorter is better).
- *Cost:* Number of blocks used to construct the vehicle (fewer is better).
```

## H.1.2. MEDIUM (Lv.2)

```
**Constraints:**
- Use only one sub-structure.
- The cargo will not show in the building process, do not include it in the building plan.

**Goal:**
- Move a 2.5 × 2.5 × 1.5 cargo with 50 units mass from the starting position (x=0, y=0) on
    the ground to the target position (x=10, y=10) on the ground (north-east direction)
   in the simulation environment.

**Evaluation Protocol:**
- The machine will be placed at (x=0, y=0) on the ground in the simulation environment.
- The cargo will be loaded to the machine by freely dropping from above the starting
   position (x=0, y=0, z=3.5).
- The cargo will not have solid connection with the machine.
- An open-loop control sequence will be programmed by a specialized AI agent following
   your plan, consisting of a list of commands with the format:
- [time: when to press the control key, command: the control key to press, duration: how
   long to hold the key]
- The trajectory of both cargo and machine will be recorded as feedback and optimized over
    three trials by adjusting the control sequence.
- The final score will be the best score across the three trials.

**Scoring Metrics:**
- *Trajectory Deviation:* Distance between the actual trajectory of the cargo and the
   ideal straight-line path from start to target (smaller is better).
- *Structure Stability:* Whether the machine remains intact during driving (higher
   stability is better).
- *Time Efficiency:* Time taken to reach the target position (shorter is better).
- *Cost:* Number of blocks used to construct the machine (fewer is better).
```

## H.1.3. HARD (Lv.3)

```
**Constraints:**
- Use only one sub-structure.
- The cargo will not show in the building process, do not include it in the building plan.

**Goal:**
```

```
- Move a 4 × 8 × 1.5 cargo (long axis along the north-south direction) with 50 units mass
   from the starting position (x=0, y=0) on the ground to the target position (x=10, y
   =10) on the ground (north-east direction), and back to the starting position in the
   simulation environment.

**Evaluation Protocol:**
- The machine will be placed at (x=0, y=0) on the ground in the simulation environment.
- The cargo will be loaded to the machine by freely dropping from above the starting
   position (x=0, y=0, z=3.5).
- The cargo will not have solid connection with the machine.
- An open-loop control sequence will be programmed by a specialized AI agent following
   your plan, consisting of a list of commands with the format:
- [time: when to press the control key, command: the control key to press, duration: how
   long to hold the key]
- The trajectory of both cargo and machine will be recorded as feedback and optimized over
    three trials by adjusting the control sequence.
- The final score will be the best score across the three trials.

**Scoring Metrics:**
- *Trajectory Deviation:* Distance between the actual trajectory of the cargo and the
   ideal straight-line path from start to target (smaller is better).
- *Structure Stability:* Whether the machine remains intact during driving (higher
   stability is better).
- *Time Efficiency:* Time taken to reach the target position (shorter is better).
- *Cost:* Number of blocks used to construct the machine (fewer is better).
```

## H.2. Support

### H.2.1. EASY (Lv.1)

```
**Constraints:**
- Use only one sub-structure.

**Goal:**
- Build a bridge capable of spanning a gap between two flat terrains (5 units wide, 5
   units high).
- The bridge must be able to support a 2.5 × 2.5 × 1.5 cargo placed at its center.

**Evaluation Protocol:**
- The terrains are positioned with edges at (x=0, y=2.5, z=5) and (x=0, y=-2.5, z=5),
   forming a 5-unit-wide gap along the north-south axis with a vertical drop of 5 units.
- The bridge will be initially placed at (x=0, y=0, z=7), slightly above the terrain tops,
    so it can gently fall into position.
- There will be no fixed connection between the bridge and the terrain.
- A cargo of size 2.5 × 2.5 × 1.5 will be dropped at (x=0, y=0, z=7), directly above the
   center of the gap.
- The cargo will rest on the bridge without any fixed connection.
- The cargo's weight will gradually and linearly increase from zero (no initial impact).
- The trajectory of the cargo will be tracked; the load at which the cargo sinks below the
    gap will be recorded as the bridge's maximum supported load.
- If the bridge fails to span the gap or misses the cargo at the start, the score is 0.

**Scoring Metrics:**
- *Maximum Load:* Maximum load supported before the cargo falls below the gap (higher is
   better).
- *Cost:* Number of blocks used to build the bridge (fewer is better).
```

### H.2.2. MEDIUM (Lv.2)

```
**Constraints:**
- Use no more than 3 sub-structures.

**Goal:**
```

```
- Build a bridge capable of spanning a gap between two flat terrains (10 units wide, 5
    units high).
- The bridge must be able to support a 2.5 × 2.5 × 1.5 cargo placed at its center.

**Evaluation Protocol:**
- The terrains are positioned with edges at (x=0, y=5, z=5) and (x=0, y=-5, z=5), forming
    a 10-unit-wide gap along the north-south axis with a vertical drop of 5 units.
- The bridge will be initially placed at (x=0, y=0, z=7), slightly above the terrain tops,
     so it can gently fall into position.
- There will be no fixed connection between the bridge and the terrain.
- A cargo of size 2.5 × 2.5 × 1.5 will be dropped at (x=0, y=0, z=7), directly above the
    center of the gap.
- The cargo will rest on the bridge without any fixed connection.
- The cargo's weight will gradually and linearly increase from zero (no initial impact).
- The trajectory of the cargo will be tracked; the load at which the cargo sinks below the
     gap will be recorded as the bridge's maximum supported load.
- If the bridge fails to span the gap or misses the cargo at the start, the score is 0.

**Scoring Metrics:**
- *Maximum Load:* Maximum load supported before the cargo falls below the gap (higher is
    better).
- *Cost:* Number of blocks used to build the bridge (fewer is better).
```

### H.2.3. HARD (Lv.3)

```
**Constraints:**
- Use no more than 3 sub-structures.

**Goal:**
- Build a bridge capable of spanning a gap between two flat terrains (20 units wide, 5
    units high).
- The bridge must be able to support a 2.5 × 2.5 × 1.5 cargo placed at its center.

**Evaluation Protocol:**
- The terrains are positioned with edges at (x=0, y=10, z=5) and (x=0, y=-10, z=5),
    forming a 20-unit-wide gap along the north-south axis with a vertical drop of 5 units.
- The bridge will be initially placed at (x=0, y=0, z=7), slightly above the terrain tops,
     so it can gently fall into position.
- There will be no fixed connection between the bridge and the terrain.
- A cargo of size 2.5 × 2.5 × 1.5 will be dropped at (x=0, y=0, z=7), directly above the
    center of the gap.
- The cargo will rest on the bridge without any fixed connection.
- The cargo's weight will gradually and linearly increase from zero (no initial impact).
- The trajectory of the cargo will be tracked; the load at which the cargo sinks below the
     gap will be recorded as the bridge's maximum supported load.
- If the bridge fails to span the gap or misses the cargo at the start, the score is 0.

**Scoring Metrics:**
- *Maximum Load:* Maximum load supported before the cargo falls below the gap (higher is
    better).
- *Cost:* Number of blocks used to build the bridge (fewer is better).
```

### H.3. `Lift`

### H.3.1. EASY (Lv.1)

```
**Constraints:**
- Use only one sub-structure.

**Goal:**
- Build a single rocket engine capable of providing propulsion to a single direction.

**Evaluation Protocol:**
```

```
- The rocket engine will be placed at position (x=0, y=0, z=0) on the ground plane.
- During the simulation, the firing control key of the rocket engine will be pressed and
    held continuously.
- The vertical propulsion force of the rocket engine will be calculated by the difference
    in vertical position of the rocket engine between the start and end of the simulation.

**Scoring Metrics:**
- *Maximum Propulsion Force:* The maximum propulsion force achieved by the rocket engine (
    higher is better).
- *Cost:* The total number of blocks used to construct the rocket engine (fewer is better)
    .
```

### H.3.2. MEDIUM (LV.2)

```
**Constraints:**
- Use only one sub-structure.

**Goal:**
- Build a rocket capable of lifting off from the ground and ascending into the sky in the
    simulation environment.

**Evaluation Protocol:**
- The rocket will be placed at position (x=0, y=0, z=0) on the ground plane.
- During the simulation, the firing control key of the rocket engine will be pressed and
    held continuously.
- The motion trajectory of the rocket will be recorded throughout the simulation.

**Scoring Metrics:**
- *Maximum Height:* The highest vertical position (z) reached by any block of the rocket (
    higher is better).
- *Trajectory Deviation:* The average lateral distance between the rocket's actual
    trajectory and the ideal vertical line (smaller is better).
- *Maximum Speed:* The highest speed achieved by any block of the rocket (higher is better
    ).
- *Cost:* The total number of blocks used to construct the rocket (fewer is better).
```

### H.3.3. HARD (LV.3)

```
**Constraints:**
- Use only two sub-structures.

**Goal:**
- Build a single rocket engine capable of providing propulsion to a single direction.
- Build a simple chassis to assemble the rocket engines using braces to form a symmetric
    rocket.
- The assembled rocket should be able to lift off from the ground to the sky in the
    simulation environment.

**Evaluation Protocol:**
- The assembled rocket will be placed at position (x=0, y=0, z=0) on the ground plane.
- During the simulation, the firing control key of the rocket engine will be pressed and
    held continuously.
- The motion trajectory of the assembled rocket will be recorded throughout the simulation
    .

**Scoring Metrics:**
- *Maximum Height:* The highest vertical position (z) reached by any block of the
    assembled rocket (higher is better).
- *Trajectory Deviation:* The average lateral distance between the assembled rocket's
    actual trajectory and the ideal vertical line (smaller is better).
- *Maximum Speed:* The highest speed achieved by any block of the assembled rocket (higher
     is better).
```

```
- *Cost:* The total number of blocks used to construct the assembled rocket (fewer is
    better).
```

# I. Workflow and Prompts

Prompts of entities in the workflow are listed as follows. `Planner`:

```
You are a functional structure building planner for a simulated build environment.
Your task is to create a detailed plan for constructing a structure that fulfills a given
    target goal.
You will be provided with a goal and a list of available building blocks.
Your plan should include an overall structure design and a breakdown of this structure
    into basic sub-structures if specified.
All sub-structures should be able to be parallel built using the available building blocks
    .

Here are the available building blocks you can use:
<available_blocks>
{available_blks}
</available_blocks>

- There will always be a default 1x1x1 shaped cubic stone starting block with weight of
    0.25 units as the base of each individual building process for each sub-structure.
- This block can't be removed, used as new block or replaced, so make sure your plan for
    each sub-structure includes the base block.

- The global coordinates of the simulation environment in [x, y, z] format are defined as:
  positive x points east,
  positive y points north,
  positive z points upward (sky).

Analyze the goal carefully and conceptualize a structure that can achieve this goal.
    Consider how the available blocks can be used to create this structure. Think about
    the physics and mechanics involved in achieving the goal.

Plan your structure by following these steps:
1. Envision an overall structure that can achieve the goal.
2. If necessary, break down this structure into non-redundant and reusable basic sub-
    structures or components, each sub-structure should be constructed independently, and
    the final structure will be assembled by attaching or connecting the sub-structures
    together.
3. For each sub-structure, determine which building blocks will be used and how they will
    be arranged.
4. Consider how these sub-structures will be assembled to form the complete structure.
5. Think about how the complete structure will function to achieve the goal.
6. Carefully compute the physical dimensions of the building blocks and the overall
    structure to ensure the structure is feasible without any overlap or conflict.
7. The structures are mainly constructed by attaching a new block to the center of an un-
    occupied face of an existing block, so you should consider the relative position of
    the new block to the existing block.
8. The attachment itself already has a connection with certain strength, brace is not
    necessary for the attachment, its only used to enhance the connection between two
    blocks that are already connected together, or to assemble structures that are not
    connected.

Your final output should be structured in the following format:

<building_plan>
<overall_structure>
  <description>
    [Provide a detailed description of the overall structure]
  </description>
  <functionality>
```

```
      [Explain how this structure works to achieve the target goal]
    </functionality>
    <assembly>
      [Describe how the sub-structures are assembled to form the complete structure if
          multiple sub-structures are specified]
    </assembly>
    <motion_control>
      [Describe the motion control and the expected motion behavior of the structure to
          achieve the target goal if the structure is expected to move]
    </motion_control>
</overall_structure>

<sub_structures>
  [For each sub-structure, include the following]
  <sub_structure_[number]>
    <name>[Name of the sub-structure]</name>
    <description>[Conceptual description of the sub-structure]</description>
    <components>[List of building blocks used]</components>
    <assembly>[How the components are arranged in the final structure if multiple sub-
        structures are specified]</assembly>
    <motion_control>[The expected motion control of the sub-structure to achieve the
        target goal if the sub-structure is expected to move]</motion_control>
    <function>[The role this sub-structure plays in achieving the overall goal]</function>
    <design_requirements>[Overall design requirements for this sub-structure]</
        design_requirements>
  </sub_structure_[number]>
  [Repeat for each sub-structure]
</sub_structures>
</building_plan>

Remember, your final output should only include the content within the <building_plan>
    tags.
Ensure that your plan is detailed, logical, and clearly explains how the proposed
    structure will achieve the given goal using the available building blocks.

- Feasibility over optimality. Produce any workable plan; do not optimize part count or
    steps unless specified.
- Explicitly include in 'design_requirements': "Positions may be micro-adjusted in later
    stages to resolve conflicts based on actual build execution."
```

Drafter:

```
You are a Drafter who designs detailed blueprints of provided machine descriptions
    following these requirements:

  - The global coordinates in [x, y, z] format are defined as:
    positive x points east,
    positive y points north,
    positive z points upward (sky).

  - There will be a default 1x1x1 cubic starting block as the base at the beginning.
    There are {available_blks} blocks available. You must only use these blocks.

  - Provide a detailed illustration of the machine meeting the requirements. You MUST
      declare all blocks in your design, and you MUST follow the given format.

  - For **static blocks** (blocks without motion or non-structural functions), describe
      placement **relative to the previous block** using compass faces (e.g., north, south
      , top, bottom).
    Format:
    '<block i> - <block type> - <block note: a brief description of the block> - <relative
        position: which face (compass) of the previous block>'

  - For **functional blocks** (blocks with motion or structural functions), provide extra
      information to describe the function and motion behavior of the block (e.g. a wheel
```

```
        that rolls towards the north, a cannon that shoots towards the south).
      Format:
      '<block i> - <block type> - <block note: a brief description of the block> - <relative
          position: which face (compass) of the previous block> - <function and motion
        behavior>'

  - The machine is constructed by placing each new block at the center of an unoccupied
      face of an existing block.

  - The coordinates of the blocks can be adjusted but mainly determined by the previous
      block.

  - You may argue with the reviewer for better solutions.

  - Your job is to translate the planner's plan into a buildable blueprint.
  - You may make position adjustments when the reviewer flags potential overlaps or when
      later build execution reveals conflicts.
  - Whenever you adjust, include a short **position adjustment note** describing what
      moved and why (e.g., "offset front axle +1 on X to clear chassis"), with flexibility
        **as needed per actual build execution**.
  - Do not change functional intent.
```

Reviewer:

```
You are a Reviewer who reviews blueprints of provided machine descriptions following these
    strict requirements:

STRUCTURAL REQUIREMENTS:

- The blueprint will be used to build the machine, so make sure the design is feasible and
    logical.
- There will be a default 1x1x1 shaped cubic starting block as the base at the beginning
    of the building process. Make sure the design has considered the base block.
- There are {available_blks}.
- For **each new block**, compute, check, and report:
1. The exact position (center coordinates) of the new block relative to the base block.
2. The distances between this new block's center and the centers of **all neighboring
    blocks** (blocks that have potential overlapping risks with the new block).
3. Whether any distance violates the minimum required distance (sum of half the block
    dimensions along the relevant axes).
- Any overlap or improper attachment must be flagged explicitly.

FUNCTIONAL VALIDATION:

- Check each point in detail, reasoning logically before proceeding to the next. Respond
    clearly whether the design meets or fails the requirement, and why.
1. Verify that the described structure allows the specified motion (e.g., rotation,
    translation). State any missing or conflicting information that prevents confirmation.
2. For all functional components (e.g., wheels, cannon, etc.), carefully calculate their
    parameters (e.g., direction of motion, direction of shooting, etc.) and validate that
    they satisfy the functional requirements specified in the description (e.g., axis
    alignment, motion direction).
3. Verify moving components have appropriate mounting and alignment. Make sure their
    mounting and alignment are consistent with the expected motion behavior.

REVIEW PROCESS:

- First, **systematically check structural integrity and collision-free placement one
    block at a time** as outlined above.
- Then, validate functional implementation.
- Finally, assess physical feasibility.
- Only approve designs that pass all three checks.

Your review should present your analysis clearly in **step-by-step format**, showing your
    calculations and reasoning for each block.
```

```
If you believe the latest version of the blueprint has fully met the design requirements,
    please give your analysis to support this belief and include 'TERMINATE' in your reply
     to finish the process.

- Prioritize feasibility over optimality. Check for overlaps, structural/functional
    conflicts, and ambiguous placement.
- When the design is acceptable, reply with 'TERMINATE'. Otherwise, be specific about
    which placements likely collide or are under-constrained.
```

Builder:

```
You are an engineering building assistant server specialized in building functional
    structures in a simulated build environment following these requirements:

- You will be equipped with a series of tools to build the structure, and your role is to
    carefully follow the instruction from your collaborator, use suitable tools to fullfil
     the instruction, make suggestions of your tool during the conversation to help to
    accomplish the requirement of the collaborator.

- You MUST NOT make parallel tool calls, you can only make one tool call in your reply.
    Build the structure one block at a time.

- The simulation environment is a 3D space with a global coordinate system in [x, y, z]
    format, where positive x points east, positive y points north, and positive z points
    upward (sky).

- **IMPORTANT**: Start the conversation with a detailed introduction of all your tools,
    describe what they can accomplish, and what information you need to fully utilize them
    .

- Be sure to mention that the note argument of some tools is very important and useful to
    mark down the specific function of the block as a powerful identifier for the block.

- Execute guidance instructions step by step. Do not infer missing intent or change parts.
```

Guidance:

```
You are an engineering building engineer who gives step by step building instructions to
    build a functional structure in a simulated build environment:

- There are {available_blks}.
- The global coordinates of the simulation environment in [x, y, z] format are defined as:
    positive x points east,
    positive y points north,
    positive z points upward (sky).
- You will be provided with a design blueprint and a description of its functionality,
    your task is to determine the detailed building steps based on the blueprint.
- Make only one move in each reply to build the structure step by step, after the
    instruction is executed by the builder, you should analyze the latest structure
    feedback from the simulation environment, and decide the next step.
- If the execution of the instruction fails, you are encouraged to acquire the necessary
    information to analyze the failure, and give the next step instruction to correct the
    process.
- The building of the structure is mainly conducted by attaching a new block to an
    unoccupied face of an existing block, but you can also use other tools to adjust the
    structure if necessary.
- Ask the builder if you have any unclear information about the permitted building
    operations/tools.
- Do not be fully restricted to the blueprint, you can make some adjustment to the
    structure as long as it meets the design requirement and the structure can function as
     intended.
- There will be a default 1x1x1 shaped cubic starting block as the base at the beginning.
    This builder shall start the building process by initialize this block once the
```

```
     instruction is given.
- After you give the final step instruction, do not end the conversation yet, you MUST
     send the requirement to review the full structure at least once to make sure the final
      building process has been executed and the structure has been updated successfully.
- If you believe the latest structure is consistent to the blueprint, please give your
     analysis to support this belief and include 'TERMINATE' in your reply to finish the
     process.

- You may make position adjustments during execution to resolve real collisions/
     constraints uncovered at build time, keeping functional intent intact.
- If repeated attempts still fail, use the available tool to **reject the current draft /
     request redesign**. Do this only after multiple good-faith tries.
- **IMPORTANT**: DO NOT make building related tools calls in your reply, your task is to
     give detailed step by step text instructions to the builder, the builder will execute
     the operations.
```

Controller:

```
You are a control engineer. Your job is to design control configurations and control
     sequences for a machine that will be tested in a simulation environment.
Your design must fulfill the given purpose while strictly following the task's evaluation
     protocol within 30 seconds.
The avaliable blocks in the simulation environment are:
<available_blocks>
{available_blks}
</available_blocks>

# Deliverable:

Return only one JSON object wrapped in a Markdown code block with the language tag json.
     Do not include any extra commentary before or after the code block.

'''json
JSON_CONTENT
'''

## control_design: string
A detailed analysis of the machine's structure and functionality according to the task's
     evaluation protocol. Explain how you will control the machine to fulfill the purpose,
     including assumptions, key constraints, and failure modes to avoid.

## control_config: list of objects
- Each object binds one key to one action on a specific block:

- key: string – must be one of:

  "UpArrow", "DownArrow", "LeftArrow", "RightArrow",

  "Alpha#" where # is 0-9 (e.g., "Alpha0", "Alpha7"),

  "Keypad#" where # is 0-9 (e.g., "Keypad3").

- action: string – the action you want this key to trigger, it MUST be one of the actions
     listed in the machine summary.

- block_id: string | integer – the identifier of the block the action applies to.

## control_sequence: list of objects
- Each object schedules a command on the timeline:

 - motion_action: string – a clear, detailed description of the commanded behavior, its
     purpose, and how it is implemented (should reference a key/action from control_config
     ).
```

```
 - time: number - simulation time in seconds when the key is pressed. Must be >= 0. Use
     floating-point if needed.

 - key: string - must be a key defined in control_config.

 - hold_for: number - how long to hold the key in seconds. Must be > 0.
```

# Rules × Constraints

## Control configurations

- You bind keys to actions on powered blocks.

- The same key may control multiple actions simultaneously (across one or more blocks).
  For example, the key "Alpha1" may control the action "spinning_forward" of block 1 and
  the action "spinning_backward" of block 2 at the same time.

- A given action on a given block can also be controlled by multiple keys. For example,
  the action "spinning_forward" of block 1 can be controlled by the keys "Alpha1" and "
  Alpha2" at the same time.

## Control sequences

- Adding a sequence entry means: at time, press key, hold it for hold_for to activate the
  bound actions, then release it. For example, the sequence entry { "time": 1.0, "key":
  "Alpha1", "hold_for": 1.0 } means: at 1.0 seconds, press the key "Alpha1", hold it for
  1.0 seconds to activate the bound actions, then release it.

- Sequence entries may overlap in time.

- The machine will execute pressed keys by invoking all actions bound to those keys in
  control_config.

- Sort control_sequence by ascending time. Use consistent units (seconds).

- The simulation only proceeds for 30 seconds, any actions beyond 30 seconds will be
  ignored.

## Quality Bar

- Be task-driven: tie decisions explicitly to the evaluation protocol and the purpose.

- Be specific and measurable: include thresholds, margins, and safety checks when relevant
  .

- State assumptions if required inputs are missing, but keep them realistic and minimal.

- Prefer concise technical language; avoid fluff.

- Ensure internal consistency between control_config and control_sequence (keys used in
  sequences must exist in the config; actions referenced must be bound as specified).

- Output must contain valid JSON and wrapped exactly as the following format:

```json
{ "control_design": "str, The detailed analysis of the machine's structure and
    functionality according to the evaluation protocol of the task, explain the how you
    would like to control the machine to fulfill the given purpose",
"control_config":
  [
    {
      "key": "str, The key you decide to use, it must be one of the keys ['UpArrow', '
          DownArrow', 'LeftArrow', 'RightArrow', 'Alpha#', 'Keypad#'] where # is a number
          from 0 to 9",
      "action": "str, The action you want to use for the key",
```

```
      "block_id": "str | int, The block id of which the action is applied"
    }
  ],
"control_sequence":
  [
    {
      "motion_action": "str, The detailed explanation of the action you want take, what is
          the purpose of this action and how to implement it",
      "time": "float, The time you decide to press the key",
      "key": "str, The key you decide to press, it must be one of the keys in your control
          config",
      "hold_for": "float, The duration you decide to press the key in seconds"
    }
  ]
}
```
# Control, Simulation, and Revision

- The control_config and control_sequence are the control configurations and sequences
    that guide the machine's actions.
- The simulation is the simulated motion trajectory of the machine, describing the motion
    trajectory (x, y, z) of some blocks in the machine.
- You should analyze the simulation and the control_config and control_sequence to revise
    the control_config and control_sequence to optimize the task.
```

## J. Additional Experiment Prompts

### J.1. One-shot Prompting for `Support` Lv.1

The following prompt is used in the one-shot experiment (Section 3.2.3). It appends a construction example to the standard **Support** Lv.1 task prompt. The example describes a bridge that achieves a maximum load of 1000.

```
**Constraints:**
- Use only one sub-structure.

**Goal:**
- Build a bridge capable of spanning a gap between two flat terrains (5 units wide, 5
    units high).
- The bridge must be able to support a 2.5 × 2.5 × 1.5 cargo placed at its center.

**Evaluation Protocol:**
- The terrains are positioned with edges at (x=0, y=2.5, z=5) and (x=0, y=-2.5, z=5),
    forming a 5-unit-wide gap along the north-south axis with a vertical drop of 5 units.
- The bridge will be initially placed at (x=0, y=0, z=7), slightly above the terrain tops,
    so it can gently fall into position.
- There will be no fixed connection between the bridge and the terrain.
- A cargo of size 2.5 × 2.5 × 1.5 will be dropped at (x=0, y=0, z=7), directly above the
    center of the gap.
- The cargo will rest on the bridge without any fixed connection.
- The cargo's weight will gradually and linearly increase from zero (no initial impact).
- The trajectory of the cargo will be tracked; the load at which the cargo sinks below the
    gap will be recorded as the bridge's maximum supported load.
- If the bridge fails to span the gap or misses the cargo at the start, the score is 0.

**Scoring Metrics:**
- *Maximum Load:* Maximum load supported before the cargo falls below the gap (higher is
    better).
- *Cost:* Number of blocks used to build the bridge (fewer is better).

**Example:**
- **Bridge Body Base:**
  Build the main bridge deck as a flat platform measuring 7 units long and 5 units wide.
      Starting from the starting block, extend symmetrically in the two horizontal
```

```
            directions. First, extend 2 small wooden blocks symmetrically along the width to
            form a wooden strip of total length 5. Then extend 3 rows of blocks symmetrically on
             each side along the length direction, resulting in a total of 34 wooden blocks plus
             1 starting block.

- **Guard Rails:**
  On top of this base, add 3 wooden blocks on each side along the width-direction edges
      adjacent to the central starting block, forming side guard rails.

- **Support Pillars:**
  Then, beneath the blocks located 1 unit in front of and 1 unit behind the central block
      along the length direction, build 2 support pillars, each consisting of 5 wooden
      blocks.

- **Reinforcement:**
  Finally, use steel rods to horizontally connect the seams on the top surfaces of the 5
      block columns to strengthen the structure. A total of 7 rows of connections are made
      , with 4 connections per row, for a total of 28 connections. Each connection point
      is located at the center of the top surfaces of two adjacent wooden blocks.
```

## J.2. Complex Bridge Construction Prompt

The following prompt is used in the construction complexity experiment (Figure 11). It explicitly emphasizes structural complexity, architectural style, and visual elaborateness, demonstrating that the benchmark framework supports complex constructions when task formulations demand them.

```
**Constraints:**
- Use 5 sub-structures, each sub-structure must be less than 100 blocks.
- The bridge must be designed in the style of London Tower Bridge.
- The structure must include at least two bridge towers.
- The bridge deck must be suspended using connector blocks.
- Temporary scaffolds made of blocks may be used during construction and then removed
    afterward to create hollow sections, layered forms, or height variation.

**Goal:**
- Build a highly elaborate and visually striking bridge inspired by London Tower Bridge
    using only one sub-structure.
- The design should emphasize architectural beauty, symmetry, vertical layering, and rich
    ornamental detail.
- The bridge must span a 10-unit gap between two flat terrains.
- The bridge deck must be suspended by ropes connected to the towers.
- The bridge deck must be at least 5 units above the bottom of the gap.
- The design should make extensive use of diverse block types to maximize artistic
    expression and construction complexity.

**Evaluation Protocol:**
- The terrains are positioned to form a 10-unit-wide gap along the north-south axis.
- The bridge will be initially placed above the gap so it can gently fall into position.
- There will be no fixed connection between the bridge and the terrain.
- The bridge must successfully span the full 10-unit gap using only one sub-structure.
- The final structure must include at least two distinct towers rising above the deck.
- The bridge deck must be suspended from the towers using rope elements rather than
    supported only from below.
- Temporary scaffold-based construction is allowed during assembly, and scaffold elements
    may be removed afterward to achieve hollow interiors, open frames, suspended details,
    or multiple height levels.
- The bridge deck must remain at a height of at least 5 units above the bottom of the gap.
- The structure will be evaluated for successful spanning, compliance with the Tower
    Bridge-inspired form, suspended deck design, visual richness, and overall
    architectural complexity.
- Designs that fail to span the gap, lack the required towers, fail to suspend the deck
    with ropes, or do not satisfy the deck-height requirement will receive a score of 0.

**Scoring Metrics:**
```

```
- *Structural Validity:* Whether the bridge successfully spans the 10-unit gap with a deck
    height of at least 5 units.
- *Tower Bridge Style Fidelity:* How strongly the final form resembles a London Tower
    Bridge-like design, including multiple towers and suspended deck composition.
- *Suspension Design Quality:* Effectiveness and elegance of the rope-supported bridge
    deck.
- *Aesthetic Complexity:* Degree of ornamentation, layered detail, hollow construction,
    and architectural sophistication.
- *Block Variety:* Number and diversity of block types used in the design (more is better)
    .
- *Cost:* Number of blocks used to build the bridge.
```

