# OpenReview forum: "BuildArena: A Physics-Aligned Interactive Benchmark of LLMs for Engineering Construction"
_ICML.cc/2026/Conference — ICML 2026 regular_

### Official Review · Reviewer_kRTo · 2026-03-12

**Soundness:** 2
**Presentation:** 3
**Significance:** 2
**Originality:** 3
**Overall Recommendation:** 3
**Confidence:** 3

**Summary:**

This paper introduces BuildArena, a physics-aligned interactive benchmark for evaluating LLMs on engineering construction tasks.

The benchmark defines three task categories (Transport: horizontal vehicle movement, Support: bridge load bearing, Lift: rocket vertical flight) each at three difficulty levels, yielding nine task-difficulty combinations.

A key technical contribution is an open-source Spatial Geometric Computation Library that mirrors the closed-source Besiege game's construction logic, enabling LLMs to interact with the construction space via language.

The paper also designs a multi-agent workflow with five collaborative LLM entities (Planner, Drafter, Reviewer, Builder, Guidance) to execute construction. Evaluation is conducted on nine closed-source LLMs with 64 samples per task for reliability, using Besiege's physics simulator.

Results show that GPT-5 and Grok-4 lead, but most models struggle severely at higher difficulty levels, with success rates dropping to zero on many Lv.2/Lv.3 tasks. The paper also provides failure mode analysis, cost-performance tradeoff analysis, and a six-dimensional difficulty characterization.

**Compliance With Llm Reviewing Policy:**

Affirmed.

**Final Justification:**

Weak Reject

**Key Questions For Authors:**

Have you considered expanding the task set? Even within the three categories, parametric variations (different gap widths for Support, different terrain for Transport, different payload masses for Lift) could substantially increase diversity beyond nine fixed scenarios. What prevents this, and is it planned?

How sensitive are the results to the multi-agent workflow design? The ablation in Appendix D Table 4 is a start, but a comparison with a simpler single-agent baseline (one LLM doing everything) would help isolate how much the workflow contributes versus the LLM's own reasoning.

Why are no open-source models included in the evaluation? Given that the library is open-sourced, testing models like Llama or open DeepSeek variants would make the benchmark substantially more useful to the community.

**Limitations:**

The authors honestly acknowledge two key limitations: the single-shot construction setting (no closed-loop refinement from simulator feedback in the main experiments) and the limited module library diversity.

**Strengths And Weaknesses:**

Strengths

The domain is genuinely novel and well-motivated. Physics-grounded 3D construction from natural language is an important capability for engineering automation that no prior benchmark tests (Table 1). Using Besiege's physics simulation as the evaluation environment provides realistic physical validation that goes beyond static metrics, and the open-source Spatial Geometric Computation Library is a tangible contribution enabling reproducibility.

The six-dimensional difficulty framework (Quantification, Robustness, Magnitude, Compositionality, Precision, Ambiguity) provides a principled lens for analyzing model capabilities. The radar chart analysis (Figure 7) reveals that all LLMs share a consistent strength/weakness profile (strong on Magnitude and Ambiguity, weak on the other four), which is an actionable insight.

The failure mode analysis (Figure 6) is informative. The finding that overlap conflict and face-occupied errors dominate across tasks, and that failure distributions differ by task category (excess connection in Support, misjudged face presence in Lift), provides concrete guidance for improving LLM spatial reasoning capabilities.

Weaknesses

The benchmark contains only nine task-difficulty combinations (three categories times three levels). While each is sampled 64 times for reliability, the task diversity is extremely narrow. Real engineering construction involves thousands of distinct design challenges. With only nine scenarios, it is difficult to draw generalizable conclusions about LLM construction capabilities, and the benchmark may overfit to these specific problems.

Performance is heavily entangled with the multi-agent workflow design. The five-entity pipeline (Planner, Drafter, Reviewer, Builder, Guidance) introduces many design choices that affect outcomes independently of the LLM's inherent construction reasoning. For example, the Reviewer's approval criteria, the Guidance entity's step-by-step instruction style, and the Builder's action parsing all influence success rates. The paper does not test alternative workflows systematically, making it unclear how much of the observed performance reflects LLM capability versus workflow engineering.

Only closed-source models are evaluated. No open-source or open-weight models (e.g., Llama, Qwen, DeepSeek open variants) are tested, which limits the benchmark's utility for the research community. Given that the library and code are released, this omission is surprising and significantly reduces the paper's contribution as a benchmark that others can build on.

---

> ### Author Rebuttal · Authors · 2026-03-31
>
> We thank the reviewer for the insightful comments. Below, we address the reviewer's concerns accordingly.
>
> > Comment 1: The task diversity is extremely narrow compared to real construction challenges.
> >
>
> **Reply:** We agree that the current benchmark suite is **compact,** and that it does not exhaust real-world engineering construction challenges. We therefore do **not** intend to claim that these task-difficulty combinations provide a complete evaluation of LLMs' general construction ability.
>
> BuildArena provides a **standardized and extensible initial benchmark** for construction tasks. The nine tasks are principled instantiations of our task design framework. The goal of the current suite is to provide a compact but structured coverage of distinct capability demands.
>
> In addition, each task remains highly **open-ended**: even within a fixed task specification, models can produce a wide range of valid designs with different geometries and topologies. Thus, the evaluation is not limited to nine narrow target outputs. Empirically, the current suite already exhibits meaningful discrimination across models and difficulty levels, indicating its informative feature despite limited size.
>
> > Comment 2 and 5: The paper does not test alternative workflows systematically, making it unclear how much the performance reflects LLM capability versus workflow.
> >
>
> **Reply:** We agree that the observed performance is influenced not only by the base LLM, but also by workflow-level design choices. To better assess this, we conducted an additional **single-agent vs. multi-agent ablation**. Please refer to the Reply to Reviewer HRaV, Comment 4. We treat the current workflow as a **shared evaluation protocol** rather than an optimized upper bound.
>
> > Comment 3 and 6: No open-source or open-weight models are tested.
> >
>
> **Reply:** We agree that evaluating on open models is important and benificial for the broader research community. We have now added three open-weight models and evaluated them on the full task suite (Table 3).
>
> Table 3. Average (n = 64) performance comparison on different tasks across task levels.
>
> | Task |  Model | Number of Parts Lv.1 | Number of Parts Lv.2 | Number of Parts Lv.3 | Success Rate (%) ↑ Lv.1 | Success Rate (%) ↑ Lv.2 | Success Rate (%) ↑ Lv.3 | Indicator ↑ Lv.1 | Indicator ↑ Lv.2 | Indicator ↑ Lv.3 |
> | --- | --- | --- | --- | --- | --- | --- | --- | --- | --- | --- |
> | Transport | Qwen3.5-9b | 3.5 | 4.3 | 2.4 | 0.0 | 0.0 | 0.0 | 1.5 | 0.8 | 0.5 |
> |  | Qwen3.5-27b | 7.0 | 18.3 | 48.0 | 12.5 | 3.1 | 12.5 | 14.5 | 5.1 | 8.4 |
> |  | Ministral-14b | 10.1 | 10.1 | 14.2 | 4.7 | 0.0 | 1.6 | 8.1 | 1.9 | 2.4 |
> | Support | Qwen3.5-9b | 5.4 | 0.3 | 0.0 | 3.1 | 0.0 | 0.0 | 7.8 | 0.0 | 0.0 |
> |  | Qwen3.5-27b | 23.6 | 29.1 | 61.0 | 35.9 | 10.9 | 0.0 | 166.0 | 36.1 | 0.0 |
> |  | Ministral-14b | 18.6 | 2.1 | 1.2 | 1.6 | 0.0 | 0.0 | 3.5 | 0.0 | 0.0 |
> | Lift | Qwen3.5-9b | 3.0 | 3.0 | 0.2 | 1.6 | 0.0 | 0.0 | 0.4 | 1.7 | 0.1 |
> |  | Qwen3.5-27b | 4.5 | 5.0 | 0.5 | 10.9 | 0.0 | 0.0 | 0.9 | 1.8 | 0.5 |
> |  | Ministral-14b | 5.9 | 11.4 | 0.5 | 12.5 | 0.0 | 0.0 | 0.8 | 2.7 | 0.1 |
>
> > Comment 4: Have you considered expanding the task set? Even within the three categories, parametric variations could increase diversity beyond nine fixed scenarios. What prevents this, and is it planned?
> >
>
> **Reply:** Yes, we have definitely considered expanding the task set in exactly this direction. The reason we started with the current **3 categories × 3 difficulty levels** is that we wanted a **compact, standardized initial suite** with controlled coverage and clear comparability. In this first version, our goal was to establish a nontrivial and discriminative benchmark core before introducing a much larger combinatorial family of task variants.
>
> In fact, the kinds of parameterized changes the reviewer mentions are precisely one of the extensions enabled by our task design framework, since they systematically modulate dimensions. We therefore view this as a very natural next step rather than a limitation of the framework itself.
>
> > Comment 7: The authors honestly acknowledge two key limitations: the single-shot construction setting (no closed-loop refinement from simulator feedback in the main experiments) and the limited module library diversity.
> >
>
> **Reply:** We agree that the main experiments primarily evaluate **single-shot construction**. To better address this limitation, we have added a new **closed-loop analyst agent experiment** on Lift Lv.2 task. As shown in Table 1 in the reply to Reviewer ARhv, the gains are modest but non-zero, suggesting that the benchmark can support post-execution refinement, even though robust improvement remains challenging.
>
> For the second point, we agree that the current module library is still limited. This is a design choice in the first benchmark release: our goal is to establish a standardized evaluation starting point, rather than to enumerate a highly enriched module space.

---

> > ### Author Rebuttal · Reviewer_kRTo · 2026-04-02
> >
> > thanks for the reply.

---

### Official Review · Reviewer_xTgw · 2026-03-13

**Soundness:** 2
**Presentation:** 1
**Significance:** 2
**Originality:** 2
**Overall Recommendation:** 2
**Confidence:** 3

**Summary:**

This paper introduces BuildArena, a physics-aligned benchmark for evaluating whether LLMs can convert natural-language engineering instructions into physically feasible 3D constructions. The benchmark defines three task families, i.e., Support, Transport, and Lift, with multiple difficulty levels, and its main contributions are an extensible task framework for language-driven construction, an open 3D Spatial Geometric Computation Library for executing build operations in Besiege, and an empirical evaluation pipeline that measures frontier LLMs on these tasks using simulation-based metrics.

**Compliance With Llm Reviewing Policy:**

Affirmed.

**Final Justification:**

I will maintain my score, as mentioned in my rebuttal acknowledgement.

**Key Questions For Authors:**

1. Could the authors kindly explain more details about Figure 8? For example, what does "normalized performance" mean and which metric it refers to? The x axis says "output tokens (k)", while the ticks say 100k to 800k. I suppose this is a typo? What does "mean/median" refer to, like the mean for all nine baseline considered? Besides, are the hundreds of output tokens used in one generation or multiple generations?

**Limitations:**

Yes.

**Strengths And Weaknesses:**

Strengths:
1. This paper tackles a relatively novel and underexplored question, i.e., language-driven 3D construction under physical constraints.
2. The evaluation setting considers different difficulty levels, which may facilitate a comprehensively test LLMs' reasoning capability.
3. The empirical study is fairly extensive: the paper evaluates nine frontier models and reports and analyzes the results.

Weaknesses:
1. The paper’s main contribution is somewhat unclear: the benchmark and tooling seem to be the real contribution, while the five-role agentic workflow appears more like supporting infrastructure than a novel method.
2. One of the two claimed contributions is "We develop a key framework module: a 3D Spatial Geometric Computation Library". However, in the main body of the paper there is only very limited description on the upper right corner of page 4. Even in Appendix F, where I suppose the detailed description to be located at, the narratives are rather disorganized. For example, F, F.3 and F.3.2 share nearly the same title, making it hard to digest the structure of this section. It's hard to understand why it is called a "library", what the library holds. Besides, the author only gives very primitive pseudocode, without one specific detailed example in F.3.2.
3. The experiments rely on the  BESIEGE SIMULATOR. How the proposed pipeline interacts with the simulator is not clearly stated, while it is an important technical component of this paper. Much description is spent on the authors' design of the three tasks, which are highly substitutable,  rather than technical design principles/design ideas which are the real important contribution.
4. The external validity is limited by the choice of Besiege and a restricted module library, so claims about broader engineering automation should be scoped more carefully.
5. LLMs are not specifically trained or tuned for the construction task designed by this paper. Therefore, to fully leverage the power of LLMs, one (or a few) example(s) should be provided to LLMs before testing their capability. If I understand correctly, the pipeline in this paper only provides the question to the LLMs without giving example(s) in advance. If this is true, why does the tests in this paper make sense, considering LLMs are not specifically trained or tuned for the construction task? Note that this task is not a common sense question, but a special task.
6. This paper only defines 9 different prompts (questions), so the task diversity is quite low. For a generation task, only repeatedly using 9 generation cases is an unconvincing test.

---

> ### Author Rebuttal · Authors · 2026-03-31
>
> We thank the reviewer for the valuable and detailed feedback. Below, we address the reviewer's concerns accordingly.
>
> > Comment 1: The paper’s main contribution is somewhat unclear.
> >
>
> **Reply:** To clarify, we do **not** claim the five-role workflow as the main methodological contribution of the paper. As stated in the introduction, our technical contributions are: **(1)** an extensible task design strategy with task categories, difficulty levels, and evaluation metrics, and **(2)** the 3D Spatial Geometric Computation Library. The five-role workflow is included as **a baseline controller** for exercising the benchmark under a fixed and reproducible protocol.
>
> > Comment 2: One of the two claimed contributions is a 3D Spatial Geometric Computation Library. However, in the main paper there is only very limited description.
> >
>
> **Reply:** We agree that the current presentation is insufficient. We have now added a detailed documentation to the anonymous codebase. We kept the main text short to prioritize benchmark design itself rather than implementation. The pseudocode was intended to illustrate the **tools exposed to the LLMs**, rather than the algorithm. We refer to it as a **library** since it provides a reusable collection of modules including geometric computations, state management, collision validation, and feedback generation.
>
> > Comment 3: How the proposed pipeline interacts with the simulator is not clearly stated.
> >
>
> **Reply:** After the workflow produces a structure, the library save it into the file required by the Besiege. An automation script is used to control the simulator to load the file, run the simulation, and capture the trajectory.
>
> We understand this impression that the current manuscript places more emphasis on tasks than on the technical design. However, the tasks are principled instantiations of our task design framework, which is defined along six engineering difficulty dimensions as described in Section 2.1. The three task families were selected to systematically stress these dimensions across both static and dynamic mechanics, while also providing a compact initial benchmark suite.
>
> > Comment 4: Claims about broader engineering automation should be scoped more carefully.
> >
>
> **Reply:** We agree that our current focus is strictly on engineering construction and spatial assembly. This design choice was made to establish a controlled and reproducible testbed specifically for spatial reasoning and physical constraint verification. Nevertheless, we believe BuildArena represents a initial step toward the promising domain of LLM-based engineering intelligence.
>
> > Comment 5: Currently the pipeline only provides the question to the LLMs without giving example(s).
> >
>
> **Reply:**
>
> We agree that few-shot prompting is useful for some tasks. However, we **intentionally adopt the zero-shot protocol**, since these tasks are **open-ended reasoning problems**. The desired structure must be derived through reasoning. Providing examples would reduce diversity and biasing the evaluation to imitation rather than generation.
>
> We further validated this with an additional **one-shot vs. zero-shot** comparison on **Support Lv.1**. In this setting, providing a strong construction example (acheives max load of 1000) **reduces performance** for both tested models (Table 2).
>
> Table 2:  Average (n = 64) performance comparison on Support-Lv.1 task with different prompting.
>
> | Model | Prompt | Success Rate (%) ↑ | Indicator (Max Load) ↑ |
> | --- | --- | --- | --- |
> | GPT-4o | zero-shot | 40.6 | 181.2 |
> |  | one-shot | 25.0 (15.6 ↓) | 160.6 (20.6 ↓) |
> | GPT-5 | zero-shot | 85.9 | 324.9 |
> |  | one-shot | 65.6 (20.3 ↓) | 294.0 (30.9 ↓) |
>
> > Comment 6: The task diversity is quite low.
> >
>
> **Reply:** Since these tasks are **open-ended reasoning questions**: each admits a large space of valid solutions, with variation in geometry, topology, and assembly strategy. Therefore, unlike conventional generation tasks with a narrow target output, repeatedly evaluating the same task instance does not reduce to repeating generation cases. Empirically, the current task suite already exhibits discriminative power across models and levels, suggesting that it is not trivial or saturated despite its compact size.
>
> > Comment 7: Explain more details about Figure 8.
> >
>
> **Reply:** Within each tak family, we take the performance of the **best single run** as **1.0**, and normalize all other runs relative to the it to yield normalized performance. We collect all individual runs in a family, group them into bins by total output token, and compute the statistics **within each token bin**.
>
> Since the tick labels already denote the token counts in thousands, current labeling is indeed redundant. We will correct this to **"Output Tokens"**. These token counts correspond to the **total output tokens consumed in one generation**.

---

> > ### Author Rebuttal · Reviewer_xTgw · 2026-04-03
> >
> > Thank the authors for the clarifications and explanations. However, in my opinion, the paper still suffers from limited contribution to the AI community and limited presentation clarity.
> >
> > It may benifit this authors to further revise the paper and make another submission in the future.
> >
> > For the current version, I will maintain my score.

---

### Official Review · Reviewer_HRaV · 2026-03-13

**Soundness:** 4
**Presentation:** 3
**Significance:** 3
**Originality:** 3
**Overall Recommendation:** 4
**Confidence:** 4

**Summary:**

BuildArena is the first physics-aligned interactive benchmark designed to evaluate Large Language Models (LLMs) in language-driven engineering construction tasks. By leveraging the Besiege physics sandbox and a custom 3D Spatial Geometric Computation Library, the framework enables LLMs to translate natural language instructions into physically viable assemblies while continuously verifying geometric and physical constraints. The system utilizes a five-role multi-agent workflow—consisting of a Planner, Drafter, Reviewer, Builder, and Guidance—to manage complex, step-by-step construction processes across three main task categories: Transport, Support, and Lift. Evaluations of nine frontier LLMs reveal that while top-tier models like GPT-5 and Grok-4 demonstrate elementary construction capabilities, success rates decline sharply as tasks increase in complexity or require higher precision. Furthermore, the study indicates that massive inference costs do not guarantee superior performance, as failed attempts often consume significantly more tokens than successful designs

**Compliance With Llm Reviewing Policy:**

Affirmed.

**Key Questions For Authors:**

As above, could the authors try to explore whether model capabilities could be enhanced through some learning approaches. For instance, one could train a smaller, specialized model or allow agents to autonomously explore the environment to accumulate modular 'skills'. Proposing these learning methodologies such as RL or iterative skill acquisition could provide a clear roadmap for improving performance on these challenging benchmarks.

**Limitations:**

As an engineering-centric project, the paper lacks a rigorous quantitative analysis of how individual components, such as the translator's fidelity relative to the original simulator's closed-source logic, affect the final evaluation results. Furthermore, the efficiency and performance of the agentic workflow are not deeply analyzed across all tiers, leaving it unclear whether the near-zero success rates in Lv.3 tasks stem from model reasoning limitations or errors inherent in the translator and multi-agent pipeline.

**Strengths And Weaknesses:**

**Strengths**:
- The paper introduces a compelling task by measuring LLM capabilities through the simulated construction of real-world infrastructure, such as bridges, vehicles, and rockets.
- The authors developed a 3D Spatial Geometric Computation Library and a multi-agent workflow that work together to turn natural language instructions into precise, physically aligned construction actions.
- The study provides a comprehensive evaluation of nine frontier models using 64 trials per task to ensure statistical reliability, while offering detailed insights into success rates, specific failure modes, and cost-performance trade-offs.

**Weaknesses**:
- The benchmark's reliability hinges on the fidelity of the 3D Spatial Geometric Computation Library in mirroring the closed-source Besiege logic. However, the paper lacks a quantitative evaluation of the library's own accuracy. Since 'Hard' tasks require extreme precision, any minor discrepancy between the library's conflict detection and the simulator's actual physics could lead to 'false negatives,' where valid model reasoning is rejected by the translator's imperfect internal logic.
- Since the authors have constructed a verifiable environment, it is worth exploring whether model capabilities could be enhanced through some learning approaches. For instance, one could train a smaller, specialized model or allow agents to autonomously explore the environment to accumulate modular 'skills'. Proposing these learning methodologies such as RL or iterative skill acquisition could provide a clear roadmap for improving performance on these challenging benchmarks.

---

> ### Author Rebuttal · Authors · 2026-03-31
>
> We thank the reviewer for the insightful questions, and our responses are as follows.
>
> > Comment 1: The paper lacks a quantitative evaluation of the library's own accuracy.
> >
>
> **Reply:** To validate the fidelity of the library, we manually reconstructed a 49-block machine in Besiege game following the exactly same LLM-generated action sequence, saved the result, and compared it with the one produced by the library at the block level. The errors in between are extremely small: max 1.41e-06, mean 9.14e-07 unit length of position error; and max 2.43e-05°, mean 1.38e-05° of orientation error. We therefore did not observe evidence that library inaccuracies cause false negatives.
>
> > Comment 2 and 3: It is worth exploring whether model capabilities could be enhanced through some learning approaches. Proposing learning methodologies such as RL or iterative skill acquisition could provide a clear roadmap for improving performance on these benchmarks.
> >
>
> **Reply:** We appreciate this suggestion and agree that this environment is well-suited for future learning-based extensions. These directions are indeed valuable, especially because the benchmark provides both process feedback and objective success signals.
>
> However, they are not the focus of the current manuscript. Our goal here is to evaluate the **existing capabilities of general-purpose LLMs** for language-driven and physics-grounded construction. In contrast, RL or specialized post-training would modify model parameters and shift the research question toward optimizing agents for this benchmark.
>
> To still probe the potential for iterative improvement without changing model weights, we added a closed-loop analyst agent experiment that uses simulator feedback and trajectory analysis to generate next-round improvement suggestions. As shown in Table 1 in the reply to Reviewer ARhv, the gains are modest but meaningful: all three models achieve non-zero success in later rounds. These results suggest that post-failure reflection is feasible and can sometimes rescue unsuccessful policies, although robust reflective learning remains an open challenge.
>
> > Comment 4: As an engineering-centric project, the paper lacks a rigorous quantitative analysis of how individual components, such as the translator's fidelity relative to the original simulator's logic, affect the final evaluation results. Furthermore, the efficiency and performance of the agentic workflow are not deeply analyzed, leaving it unclear whether those fails in Lv.3 tasks stem from model reasoning limitations or errors inherent in the translator and multi-agent pipeline.
> >
>
> **Reply:** Regarding the translator, as noted in our response to Comment 1, the observed discrepancies are extremely small, suggesting that translator-side geometric mismatch is unlikely to be the dominant source of failure.
>
> To analyze the workflow itself, we additionally conducted a **single-agent vs. multi-agent ablation on the three Lv.2 tasks using 2 of the models that give near zero success**. The results are mixed: the multi-agent workflow helps in some settings but hurts in others, and in several cases the single-agent setup performs better. This indicates that the current workflow is not uniformly optimal, and that its effectiveness depends on task and model. Therefore, we use it as a **fixed, shared, and reproducible evaluation protocol** so that all models are compared under the same agentic setting, without model-specific workflow tuning or prompt engineering.
>
> Importantly, under this unified protocol, the benchmark exhibits meaningful discriminative power across models. Our contribution is to provide a benchmark that can reveal capability differences under a common evaluation setting, rather than to prescribe a optimal pipeline design.

---

> > ### Author Rebuttal · Reviewer_HRaV · 2026-04-06
> >
> > Thanks for the rebuttal. Most of my concerns have been resolved. Considering my previous positive score, I would keep my rating unchanged.

---

### Official Review · Reviewer_ARhv · 2026-03-13

**Soundness:** 3
**Presentation:** 3
**Significance:** 3
**Originality:** 4
**Overall Recommendation:** 4
**Confidence:** 3

**Summary:**

This paper proposes a benchmark for evaluating the ability of LLMs to build 3D models. The main question the authors aim to answer is whether current large language models can transform natural language requirements into executable and functional 3D constructions under physical constraints. To this end, the paper presents a three-part framework: task definition, an LLM-driven construction pipeline, and simulation-based evaluation. It also introduces two main technical components: a scalable task design strategy and a 3D Spatial Geometric Computation Library that maps language instructions to construction operations in Besiege.
The experimental results suggest that current LLMs do possess some construction ability, but their overall performance is still only moderate, and they remain significantly limited on fine-grained 3D construction tasks. However, this work is built on Besiege, and the rules and object properties in the game are not representative of common 3D object properties in general. In my opinion, this makes it difficult for the benchmark to serve as a fully comprehensive evaluation of LLMs’ capabilities in 3D understanding and generation. Nevertheless, the core idea of the paper is very interesting and deserves appreciation.

**Compliance With Llm Reviewing Policy:**

Affirmed.

**Ethical Review Flag:**

Flag this paper for an ethics review.

**Final Justification:**

I think the rebuttal solve my questions.

**Key Questions For Authors:**

The current tasks seem to be relatively short-horizon. Would it be possible to evaluate longer-horizon compositional tasks as well?

And I think this paper is quite interesting. If authors can solve the weakness and question, I am glad to raise the score.

**Limitations:**

yes

**Strengths And Weaknesses:**

Strengths
1. The work uses Besiege, a simulation-like game platform, to push the problem toward the setting of “language-driven + 3D construction + physical executability,” which is indeed closer to the real demands of embodied intelligence and engineering automation, and provides a unique perspective for evaluating the 3D understanding and generation abilities of LLMs.
2. The process proposed by the authors for mapping LLM outputs to the building of 3D objects could potentially be further generalized and extended.
3. Using a game engine and gameplay mechanics to validate usability is both reasonable and highly interesting.

Weaknesses
1. I looked at some Besiege gameplay videos, and it seems that the modules used by the authors are still relatively simple in terms of the resulting constructions. As shown in Figure 9 of the appendix, could the benchmark support more modules to generate more complex objects?
2. The entire environment is built upon the module space and physical logic of Besiege. While this is convenient for interactive construction, it still remains noticeably different from real robots and real-world manufacturing constraints.
3. Some of the metric definitions are debatable. For example, the authors treat “the fewer the number of parts, the better” as a sign of engineering simplicity. While this may be reasonable in some cases, it may also overly penalize designs that use more parts but are more robust or provide greater redundancy and safety. In other words, the metrics implicitly encode the authors’ value judgment of what constitutes a “good design,” and such a judgment may not apply to all engineering tasks. This is a point that can reasonably be questioned in a benchmark paper.
4. Since the tasks ultimately require execution, the current feedback mechanism seems to operate only during the construction process. There does not appear to be a sufficiently developed process for reflection, learning, and summarization after failure in simulation execution. I did notice that this aspect is discussed in the appendix, but it should be supported with more experiments and more agent-style design.

---

> ### Author Rebuttal · Authors · 2026-03-31
>
> We thank the reviewer for the constructive comments. Below we clarify the main points accordingly.
>
> > Comment 1: Compareed to some Besiege gameplay videos, seems that the modules used by the authors are still relatively simple. Could the benchmark support more modules to generate more complex objects?
> >
>
> **Reply:** We conducted an additional qualitative study with a task explicitly emphasizing complexity, and observed more elaborate results uisng the same module set (see https://anonymous.4open.science/r/BuildArena-71DF/asset/new_results/bridge.gif). This suggests that the relative simplicity of current outputs is driven more by task formulation and LLM design than by the benchmark itself. The framework is also extensible, and additional modules can be incorporated.
>
> > Comment 2: While the Besiege environment is convenient for interactive construction, it still remains different from real robots and real-world manufacturing constraints.
> >
>
> **Reply:** We agree that a gap exists between Besiege and real-world robotics or manufacturing. However, BuildArena is designed to evaluate spatial reasoning and multi-step planning, rather than real-world embodied execution. The key question is whether a model can interpret a goal, design a structure, execute the construction, while satisfying physics constrains. This evaluation logic does not require full real-world realism.
>
> Besiege is a suitable testbed because it provides gravity, friction, collision, and joint constraints within a mature physics engine, which is sufficient to separate feasible from infeasible solutions. Our claim is that BuildArena "takes a first step towards engineering automation using LLMs," not that it can directly transfer to real manufacturing pipelines.
>
> > Comment 3: Some of the metric definitions are debatable. For example, "the fewer the number of parts, the better" as a sign of engineering simplicity.
> >
>
> **Reply:** We agree that metrics should not encode a universal value judgment about what counts as a “good” engineering design. We would like to clarify that Number of Parts **is not used for model ranking**. The ranking in Figure 7 is based on Success Rate and Indicator, followed by rank aggregation across tasks. We will make this clearer in the revision. Currently, it is reported only as a signal for construction efficiency. The prompt preference for fewer parts is meant to encourage economical solutions, but it is not enforced in evaluation.
>
> Our empirical results are consistent with the reviewer’s concern: in larger bridge-like tasks, stronger models often use more blocks for reinforcement, while in lift tasks fewer blocks can be beneficial for thrust-to-weight balance.
>
> > Comment 4: There does not incorporate reflection, learning, and summarization after failure in simulation execution.
> >
>
> **Reply:** We agree that post-execution reflection should be made more explicit. In response, we conducted an additional closed-loop experiment to test whether execution failures can be analyzed and turned into improvements.
>
> On Lift Lv.2, we selected three models with 0% one-shot success rate, and introduced an analyst agent with access to code and file tools, who reviews machine structure, simulator feedback and trajectories, summarizes failure causes, and appends improvement suggestions to the next-round prompt.
>
> The gains are modest but meaningful: all three models achieve non-zero success in later rounds. These results suggest that post-failure reflection is feasible and can sometimes rescue unsuccessful policies, although robust reflective learning remains an open challenge.
>
> Table 1: Multi-turn closed-loop construction results on Lift-Lv.2 task (Note: Round 1 means no closed-loop).
>
> |  | Model | Round 1 | Round 2 | Round 3 | Round 4 | Round 5 |
> | --- | --- | --- | --- | --- | --- | --- |
> | Success Rate (%) ↑ | DeepSeek-3.1 | 0.0 | 0.0 | 1.6 | 3.1 | 0.0 |
> |  | Qwen-3 | 0.0 | 0.0 | 1.6 | 0.0 | 1.6 |
> |  | Seed-1.6 | 0.0 | 0.0 | 1.6 | 0.0 | 0.0 |
> | Indicator (Max Height) ↑ | DeepSeek-3.1 | 3.8 | 3.0 | 3.3 | 3.5 | 2.0 |
> |  | Qwen-3 | 3.1 | 2.5 | 3.0 | 1.9 | 2.7 |
> |  | Seed-1.6 | 2.6 | 1.9 | 2.6 | 1.5 | 1.4 |
>
> > Comment 5: The current tasks seem to be relatively short-horizon. Would it be possible to evaluate longer-horizon compositional tasks as well?
> >
>
> **Reply:** We agree that longer-horizon compositional tasks are important. We'd also like to clarify that the current benchmark already includes tasks with long-horizon structure. Support Lv.2/3 and Lift Lv.3 require planning substructures, constructing components, and assembling into a final solution, it takes up to hundreds of actions to build one final structure, rather than a single-step build.
>
> In addition, solving these tasks can require hundreds of building actions with iterative environment feedback during construction. Since the framework is extensible, adding more complex multi-stage tasks is a natural direction for future work.

---

> > ### Author Rebuttal · Reviewer_ARhv · 2026-04-03
> >
> > Thanks for authors rebuttal. I think this paper is quite interesting on the video games, which are played as a world simulator. I will raise my score.

---

### Decision · Program_Chairs · 2026-04-30

**Decision:**

Accept (regular)

**Comment:**

The authors introduce BuildArena, a physics-aligned LLM benchmark for engineering construction. They use a game-like simulation called Besiege, and introduce a library of skills which map language instructions to actions in the game.

It's an interesting idea, and gets at an important problem, as LLMs are often still not overly capable of spatial reasoning tasks like construction. The authors' results show how the models still suffer at accomplishing the various tasks, though it must be noted that these are game results and probably have little bearing on real-world tasks.

The reviewers raised concerns about the applicability and relevance of BESIEGE (the simulator) to the broader AI community. Task diversity is also unfortunately fairly low -- the paper would be greatly improved with more tasks.

However, this is a genuinely novel problem, and one that's potentially very interesting, involving the assembly of multiple sub-components in some cases. They tested against some open-source LLMs. showing nonzero results. The follow-up experiments on one-shot vs. zero-shot prompting were interesting, and should likely make it into the final paper -- they make a point about the reasoning challenges involved.